# Deep Statistical Solvers

**Balthazar Donon**
RTE R&D, INRIA, Université Paris-Saclay
`balthazar.donon@rte-france.com`

**Zhengying Liu**
Université Paris-Saclay, INRIA
`zhengying.liu@inria.fr`

**Wenzhuo Liu**
IRT SystemX
`wenzhuo.liu@irt-systemx.fr`

**Isabelle Guyon**
Université Paris-Saclay, INRIA, Chalearn
`guyon@chalearn.org`

**Antoine Marot**
RTE R&D
`antoine.marot@rte-france.com`

**Marc Schoenauer**
INRIA, Université Paris-Saclay
`marc.schoenauer@inria.fr`

## Abstract

This paper introduces Deep Statistical Solvers (DSS), a new class of trainable solvers for optimization problems, arising *e.g.*, from system simulations. The key idea is to learn a solver that generalizes to a given distribution of problem instances. This is achieved by directly using as loss the objective function of the problem, as opposed to most previous Machine Learning based approaches, which mimic the solutions attained by an existing solver. Though both types of approaches outperform classical solvers with respect to speed for a given accuracy, a distinctive advantage of DSS is that they can be trained without a training set of sample solutions. Focusing on use cases of systems of interacting and interchangeable entities (*e.g.* molecular dynamics, power systems, discretized PDEs), the proposed approach is instantiated within a class of Graph Neural Networks. Under sufficient conditions, we prove that the corresponding set of functions contains approximations to any arbitrary precision of the actual solution of the optimization problem. The proposed approach is experimentally validated on large linear problems, demonstrating super-generalisation properties; And on AC power grid simulations, on which the predictions of the trained model have a correlation higher than $99.99\%$ with the outputs of the classical Newton-Raphson method (known for its accuracy), while being 2 to 3 orders of magnitude faster.

## 1 Introduction

In many domains of physics and engineering, Deep Neural Networks (DNNs) have sped up simulations and optimizations by orders of magnitude, replacing some computational bricks based on first principles with data-driven numerical models – see *e.g.*, [1, 2, 3, 4]. However, in general, such data-driven approaches consist in training a *proxy* in a supervised way, to imitate solutions provided by some numerical solver. This is sometimes infeasible due to the high computational cost of existing simulators (*e.g.*, molecular dynamics, car crash simulations, computational fluid dynamics, and power grid simulation). Furthermore, such approaches ignore problem-specific considerations and may end up providing inconsistent solutions, failing to satisfy physical laws such as energy conservation (which can only be a posteriori checked, see *e.g.* [4]). In order to bypass this weakness, a growing body of work pushes towards an interplay between physics and Machine Learning [5, 6], *e.g.*, incorporating physical knowledge in the loss function during learning [7, 8].

Another important property of natural or artificial systems is that of invariance, a fundamental concept in science, allowing to generalize conclusions drawn from few observations, to whole invariance classes. This work focuses on permutation-invariant problems, which appear in simulations of complex systems of interacting and interchangeable entities [9] (*e.g.,* molecular dynamics, power grids, simulations of partial differential equations (PDEs) with finite elements). Invariance has made its way in machine learning, as illustrated by the success of Convolutional Neural Networks (CNN) [10, 11], and of Graph Neural Networks (GNN) [12, 13]. In particular, implementations of GNNs successfully handle materials dynamics simulations [14], power systems [15], interacting particles [16] and classical [17] or quantum [18] chemistry. However, all of these works pertain to the *proxy approach* described above.

Our first contribution is to propose, at the interface of optimization and statistics, the Statistical Solver Problem (SSP), a novel formulation for learning to solve a whole class of optimization and system simulation problems. The resulting framework i) directly minimizes the global loss function of the problems during training, thus not requiring any existing solution of the problems at hand, and ii) directly incorporates permutation-invariance in the representation of the problems using a GNN-based architecture, called Deep Statistical Solver (DSS). Our second contribution is to prove that DSS satisfies some Universal Approximation property in the space of SSP solutions. The third contribution is an experimental validation of the approach.

The outline of the paper is the following. Section 2 sets the background, and defines SSPs. Section 3 introduces Deep Statistical Solvers. Section 4 proves the Universal Approximation property for permutation-invariant loss functions (and some additional hypotheses). Section 5 experimentally validates the DSS approach, demonstrating its efficiency w.r.t. state-of-the-art solvers, and unveiling some super-generalization capabilities. Section 6 concludes the paper.

## 2 Definitions and Problem Statement

This section introduces the context (notations and definitions) and the research goal of this work: The basic problem is, given a network of interacting entities (referred to later as Interaction Graph), to find a state of the network that minimizes a given loss function; From thereon, the main goal of this work is to learn a parameterized mapping that accurately and quickly computes such minimizing state for any Interaction Graph drawn from a given distribution.

### 2.1 Notations and Definitions

**Notations** Throughout this paper, for any $n \in \mathbb{N}$, $[n]$ denotes the set $\{1, \ldots, n\}$; $\Sigma_n$ is the set of permutations of $[n]$; for any $\sigma \in \Sigma_n$, any set $\Omega$ and any vector $\mathbf{x} = (x_i)_{i \in [n]} \in \Omega^n$, $\sigma \star \mathbf{x}$ is the vector $(x_{\sigma^{-1}(i)})_{i \in [n]}$; for any $\sigma \in \Sigma_n$ and any matrix $\mathbf{m} = (m_{ij})_{i,j \in [n]} \in \mathcal{M}_n(\Omega)$ (square matrices with elements in $\Omega$), $\sigma \star \mathbf{m}$ is the matrix $(m_{\sigma^{-1}(i)\sigma^{-1}(j)})_{i,j \in [n]}$.

**Interaction Graphs** We call *Interaction Graph* a system of $n \in \mathbb{N}$ interacting entities, or *nodes*, defined as $\mathbf{G} = (n, \mathbf{A}, \mathbf{B})$, where $n$ is the size of $\mathbf{G}$ (number of nodes), $\mathbf{A} = (A_{ij})_{i,j \in [n]}; A_{ij} \in \mathbb{R}^{d_A}; d_A \geq 1$ represents the interactions between nodes, and $\mathbf{B} = (B_i)_{i \in [n]}; B_i \in \mathbb{R}^{d_B}, d_B \geq 1$ are some local external inputs at each node. Let $\mathcal{G}_{d_A, d_B}$ be the set of all such Interaction Graphs and simply $\mathcal{G}$ when there is no confusion. For any $\sigma \in \Sigma_n$ and any Interaction Graph $\mathbf{G} = (n, \mathbf{A}, \mathbf{B})$, $\sigma \star \mathbf{G}$ denotes the Interaction Graph $(n, \sigma \star \mathbf{A}, \sigma \star \mathbf{B})$.

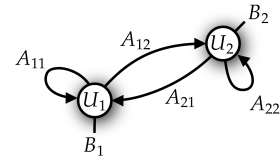

Figure 1: **A sample Interaction Graph** $(2, \mathbf{A}, \mathbf{B})$

Interaction Graphs can also be viewed as "doubly weighted" graphs, *i.e.*, graphs with weights on both the edges (weights $A_{ij}$) and the nodes (weights $B_i$), considering that those weights are vectors. For a given $\mathbf{G}$, we will also consider the underlying undirected unweighted graph $\widetilde{\mathbf{G}}$ for which links between nodes $i$ and $j$ exist *iff* either $A_{ij}$ or $A_{ji}$ is non-zero[1]. We will use the notion of neighborhood induced by $\widetilde{\mathbf{G}}$: $j \in \mathcal{N}(i; \mathbf{G})$ *iff* $i$ and $j$ are neighbors in $\widetilde{\mathbf{G}}$ (and $\mathcal{N}^\star(i; \mathbf{G})$ will denote $\mathcal{N}(i; \mathbf{G}) \backslash \{i\}$).

**States and Loss Functions** Vectors $\mathbf{U} = (U_i)_{i \in [n]}; U_i \in \mathbb{R}^{d_U}, d_U \geq 1$ represent *states* of Interaction Graphs of size $n$, where $U_i$ is the state of node $i$. $\mathcal{U}_{d_U}$ denotes the set of all such states ($\mathcal{U}$ when there

is no confusion). A *loss function* $\ell$ is a real-valued function defined on pairs $(\mathbf{U}, \mathbf{G})$, where $\mathbf{U}$ is a state of $\mathbf{G}$ (i.e., of same size).

**Permutation invariance and equivariance** A loss function on Interaction Graph $\mathbf{G}$ of size $n$ is *permutation-invariant* if for any $\sigma \in \Sigma_n$, $\ell(\sigma \star \mathbf{U}, \sigma \star \mathbf{G}) = \ell(\mathbf{U}, \mathbf{G})$.
A function $\mathcal{F}$ from $\mathcal{G}$ to $\mathcal{U}$, mapping an Interaction Graph $\mathbf{G}$ of size $n$ on one of its possible states $\mathbf{U}$ is *permutation-equivariant* if for any $\sigma \in \Sigma_n$, $\mathcal{F}(\sigma \star \mathbf{G}) = \sigma \star \mathcal{F}(\mathbf{G})$.

## 2.2  Problem Statement

**The Optimization Problem** In the remaining of the paper, $\ell$ is a loss function on Interaction Graphs $\mathbf{G} \in \mathcal{G}$ that is both continuous and permutation-invariant. The elementary question of this work is to solve the following optimization problem for a given Interaction Graph $\mathbf{G}$:

$$\mathbf{U}^\star(\mathbf{G}) = \underset{\mathbf{U} \in \mathcal{U}}{\operatorname{argmin}} \ \ell(\mathbf{U}, \mathbf{G}) \tag{1}$$

**The Statistical Learning Goal** We are not interested in solving problem (1) for just ONE Interaction Graph, but in learning a parameterized *solver*, i.e., a mapping from $\mathcal{G}$ to $\mathcal{U}$, which solves (1) for MANY Interaction Graphs, namely all Interaction Graphs $\mathbf{G}$ sampled from a given distribution $\mathcal{D}$ over $\mathcal{G}$. In particular, $\mathcal{D}$ might cover Interaction Graphs of different sizes. Let us assume additionally that $\mathcal{D}$ and $\ell$ are such that, for any $\mathbf{G} \in \operatorname{supp}(\mathcal{D})$ (the support of $\mathcal{D}$) there is a unique minimizer $\mathbf{U}^*(\mathbf{G}) \in \mathcal{U}$ of problem (1). The goal of the present work is to learn a single solver that best approximates the mapping $\mathbf{G} \mapsto \mathbf{U}^*(\mathbf{G})$ for all $\mathbf{G}$ in $\operatorname{supp}(\mathcal{D})$. More precisely, assuming a family of solvers $Solver_\theta$ parameterized by $\theta \in \Theta$ (Section 3 will introduce such a parameterized family of solvers, based on Graph Neural Networks), the problem tackled in this paper can be formulated as a *Statistical Solver Problem* (SSP):

$$\operatorname{SSP}(\mathcal{G}, \mathcal{D}, \mathcal{U}, \ell) \begin{cases} \text{Given distribution } \mathcal{D} \text{ on space of Interaction Graphs } \mathcal{G}, \text{ space of states } \mathcal{U}, \\ \text{and loss function } \ell, \text{ solve } \theta^\star = \underset{\theta \in \Theta}{\operatorname{argmin}} \ \mathbb{E}_{\mathbf{G} \sim \mathcal{D}} \ [\ell\left(Solver_\theta(\mathbf{G}), \mathbf{G}\right)] \end{cases} \tag{2}$$

**Learning phase** In practice, the expectation in (2) will be empirically computed using a finite number of Interaction Graphs sampled from $\mathcal{D}$, by directly minimizing $\ell$ (i.e., without the need for any $\mathbf{U}^\star$ solution of (1)). The result of this empirical minimization is a parameter $\widehat{\theta}$.

**Inference** The solver $Solver_{\widehat{\theta}}$ can then be used, at inference time, to compute, for any $\mathbf{G} \in supp(\mathcal{D})$, an approximation of the solution $\mathbf{U}^\star(\mathbf{G})$

$$\widehat{\mathbf{U}}(\mathbf{G}) = Solver_{\widehat{\theta}}(\mathbf{G}) \tag{3}$$

Solving problem (1) has been replaced by a simple and fast inference of the learned model $Solver_{\widehat{\theta}}$ (at the cost of a possibly expensive learning phase).

**Discussion** The SSP experimented with in Section 5.2 addresses the simulation of a Power Grid, a real-world problem for which the benefits of using the proposed approach becomes clear. Previous work [19] used a "proxy" approach, which consists in learning from known solutions of the problem, provided by a classical solver. The training phase is sketched on Figure 2.a. The drawback of such an approach is the need to gather a huge number of training examples (i.e., solutions of problem (1)), something that is practically infeasible for complex problems: either such solutions are too costly to obtain (e.g., in car crash simulations), or there is no provably optimal solution (e.g., in molecular dynamics simulations). In contrast, since the proposed approach directly trains $Solver_\theta$ by minimizing the loss $\ell$ (Figure 2.b), no such examples are needed.

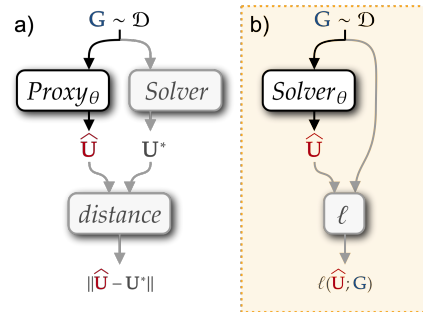

Figure 2: **Proxy approach (a)** *vs.* **DSS (b)**

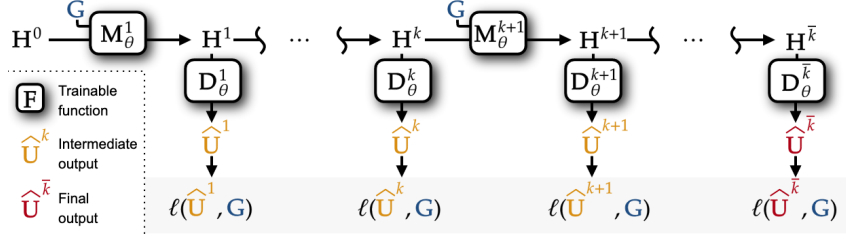

Figure 3: **Graph Neural Network implementation of a DSS**

# 3 Deep Statistical Solver Architecture

In this section, we introduce the class of Graph Neural Networks (GNNs) that will serve as DSSs. The intuition behind this choice comes from the following property (proof in Appendix B.2):

**Property 1.** *If the loss function $\ell$ is permutation-invariant and if for any $\mathbf{G} \in supp(\mathcal{D})$ there exists a unique minimizer $\mathbf{U}^*(\mathbf{G})$ of problem (1), then $\mathbf{U}^*$ is permutation-equivariant.*

Graph Neural Networks, introduced in [20], and further developed in [21, 22] (see also the recent surveys [13, 23]), are a class of parameterized permutation-equivariant functions. Therefore, they seem to be good candidates to build SSP solutions, since Property 1 states that the ideal solver $\mathbf{U}^*$ is permutation-equivariant (this will be confirmed by Corollary 1).

**Overall architecture** There are many possible implementations of GNNs, but whatever the chosen type, it is important to make room for information propagation throughout the whole network (see also Section 4). Hence the choice of an iterative process that acts on a latent state $\mathbf{H} \in \mathcal{U}_d; H_i \in \mathbb{R}^d, d \geq 1$ for $\overline{k}$ iterations ($d$ and $\overline{k}$ are hyperparameters). For a node $i \in [n]$, the latent state $H_i$ can be seen as an embedding of the actual state $U_i$.

The overall architecture is described in Figure 3. All latent states in $\mathbf{H}^0$ are initialized to a zero vector. The *message passing* step performs $\overline{k}$ updates on the latent state variable $\mathbf{H}$ using $\mathbf{M}_\theta^k$, spreading information using interaction coefficients $\mathbf{A}$ and external inputs $\mathbf{B}$ of $\mathbf{G}$ (eq. 5–8). After each update, latent state $\mathbf{H}^k$ is decoded into a meaningful actual state $\widehat{\mathbf{U}}^k$ (eq. 9). The last state $\widehat{\mathbf{U}}^{\overline{k}}$ is the actual output of the algorithm $\widehat{\mathbf{U}}$. However, in order to robustify learning, all intermediate states $\widehat{\mathbf{U}}^k$ are taken into account in the training loss through a discounted sum with hyperparameter $\gamma \in [0,1]$:

$$\text{Training Loss} = \sum_{k=1}^{\overline{k}} \gamma^{\overline{k}-k} \ell(\widehat{\mathbf{U}}^k, \mathbf{G}) \tag{4}$$

**Message passing $\mathbf{M}_\theta^k$** For each node $i$, three different messages are computed, $\phi_{\to,\theta}^k, \phi_{\leftarrow,\theta}^k, \phi_{\circlearrowleft,\theta}^k$, corresponding to outgoing, ingoing and self-loop links, respectively using trainable mappings $\Phi_{\to,\theta}^k, \Phi_{\leftarrow,\theta}^k, \Phi_{\circlearrowleft,\theta}^k$, as follows:

$$\phi_{\to,i}^k = \sum_{j \in \mathcal{N}^\star(i;\mathbf{G})} \Phi_{\to,\theta}^k(H_i^{k-1}, A_{ij}, H_j^{k-1}) \qquad \textit{outgoing edges} \tag{5}$$

$$\phi_{\leftarrow,i}^k = \sum_{j \in \mathcal{N}^\star(i;\mathbf{G})} \Phi_{\leftarrow,\theta}^k(H_i^{k-1}, A_{ji}, H_j^{k-1}) \qquad \textit{ingoing edges} \tag{6}$$

$$\phi_{\circlearrowleft,i}^k = \Phi_{\circlearrowleft,\theta}^k(H_i^{k-1}, A_{ii}) \qquad \textit{self loop} \tag{7}$$

Latent states $H_i^k$ are then computed using trainable mapping $\Psi_\theta^k$, in a ResNet-like fashion:

$$\mathbf{H}^k = \mathbf{M}_\theta^k(\mathbf{H}^{k-1}, \mathbf{G}) := (H_i^k)_{i\in[n]}, \text{ with } H_i^k = H_i^{k-1} + \Psi_\theta^k(H_i^{k-1}, B_i, \phi_{\to,i}^k, \phi_{\leftarrow,i}^k, \phi_{\circlearrowleft,i}^k) \tag{8}$$

**Decoding** The decoding step applies the same trainable mapping $\Xi_\theta^k$ to every node:

$$\widehat{\mathbf{U}}^k = \mathbf{D}_\theta^k(\mathbf{H}^k) = (\Xi_\theta^k(H_i^k))_{i\in[n]} \tag{9}$$

**Training** All trainable blocks $\Phi_{\to,\theta}^k, \Phi_{\leftarrow,\theta}^k, \Phi_{\circlearrowleft,\theta}^k$ and $\Psi_\theta^k$ for the message passing phase, and $\Xi_\theta^k$ for the decoding phase, are implemented as Neural Networks. They are all trained simultaneously, backpropagating the gradient of the training loss of eq. (4) (see details in Section 5).

**Number of propagation steps** Our current implementation choice is to consider different neural network blocks at each propagation step. The underlying intuition is that the nature of information exchange does not have to be the same at the beginning and at the end of the process. This comes at the expense of a fixed amount of propagation steps $\overline{k}$. However, future work will include the investigation of a Recurrent Graph Neural Network architecture, drawing inspiration from [24]. This would allow for an adaptive number of steps $\overline{k}$, and a much lighter model.

**Inference Complexity** Assuming that each neural network block has a single hidden layer with dimension $d$, that $d \geq d_A, d_B, d_U$, and denoting by $m$ the average neighborhood size, one inference has computational complexity of order $\mathcal{O}(mn\overline{k}d^3)$, scaling linearly with $n$. Furthermore, many problems involve very local interactions, resulting in small $m$. However, one should keep in mind that hyperparameters $\overline{k}$ and $d$ should be chosen according to the charateristics of distribution $\mathcal{D}$. If we can compute the maximal diameter of any $\mathbf{G} \in \text{supp}(\mathcal{D})$ (e.g., if $\mathcal{D}$ is a database of the history of the Californian power grid), one should choose a larger value for $\overline{k}$ (see Corollary 1). Similarly, if one is working with data that have very large $d_A$ and $d_B$, one may want to choose a sufficiently large value for $d$ to let information flow properly.

**Equivariance** The proposed architecture defines permutation-equivariant DSS (see Appendix B.1).

## 4 Deep Statistical Solvers are Universal Approximators for SSPs Solutions

This Section proves, heavily relying on work by [25], a Universal Approximation Theorem for the class of DSSs with Lipschitz activation function (*e.g.* ReLU) in the space of the solutions of SSPs. The space of Interaction Graphs is a metric space for the distance
$$d(\mathbf{G}, \mathbf{G}') = \|\mathbf{A} - \mathbf{A}'\| + \|\mathbf{B} - \mathbf{B}'\| \text{ if } n = n' \text{ and } + \infty, \text{ otherwise}$$

**Universal Approximation Property** Given metric spaces $\mathcal{X}$ and $\mathcal{Y}$, a set of continuous functions $\mathcal{H} \subset \{f : \mathcal{X} \to \mathcal{Y}\}$ is said to satisfy the *Universal Approximation Property* (UAP) if it is dense in the space of all continuous functions $\mathcal{C}(\mathcal{X}, \mathcal{Y})$ (with respect to the uniform metric).
Denote by $\mathcal{H}_{d_{in}}^{d_{out}}$ a set of neural networks from $\mathbb{R}^{d_{in}}$ to $\mathbb{R}^{d_{out}}$, for which the UAP holds. It is known since [26] that the set of neural networks with at least one hidden layer, an arbitrarily large amount of hidden neurons, and an appropriate activation function, satisfies these conditions.

**Hypothesis space** Let $\overline{\overline{k}} \in \mathbb{N}$. We denote by $\mathcal{H}^{\overline{\overline{k}}}$ the set of graph neural networks defined in Section 3 such that $\overline{k} \leq \overline{\overline{k}}$, $d \in \mathbb{N}$ and for any $k = 1, \ldots, \overline{k}$, we consider all possible $\Phi_{\to,\theta}^k, \Phi_{\leftarrow,\theta}^k \in \mathcal{H}_{d_A+2d}^d$, $\Phi_{\circlearrowleft,\theta}^k \in \mathcal{H}_{d_A+d}^d$, $\Psi_\theta^k \in \mathcal{H}_{d_B+4d}^d$ and $\Xi_\theta^k \in \mathcal{H}_d^{d_U}$.

**Diameter of an Interaction Graph** Let $\mathbf{G} = (n, \mathbf{A}, \mathbf{B}) \in \mathcal{G}$, and let $\widetilde{\mathbf{G}}$ be its undirected and unweighted graph structure, as defined in Section 2.1. We will write $\text{diam}(\mathbf{G})$ for $\text{diam}(\widetilde{\mathbf{G}})$, the diameter of $\widetilde{\mathbf{G}}$ [27].

**Hypotheses over distribution** $\mathcal{D}$ We introduce the four following hypotheses over $\text{supp}(\mathcal{D})$:
- *Permutation-invariance.* For any $\mathbf{G} \in \text{supp}(\mathcal{D})$ and $\sigma \in \Sigma_n$, $\sigma \star \mathbf{G} \in \text{supp}(\mathcal{D})$;
- *Compactness.* $\text{supp}(\mathcal{D})$ is a compact subset of $\mathcal{G}$;
- *Connectivity.* For any $\mathbf{G} \in \text{supp}(\mathcal{D})$, $\widetilde{\mathbf{G}}$ has only one connected component;
- *Separability of external inputs.* There exist $\delta > 0$ such that for any $\mathbf{G} = (n, \mathbf{A}, \mathbf{B}) \in \text{supp}(\mathcal{D})$ and any $i \neq j \in [n]$, $\|B_i - B_j\| \geq \delta$.

The *compactness* implies that there is an upper bound $\overline{n}$ over the size $n$ of Interaction Graphs in $\text{supp}(\mathcal{D})$. Also, these hypotheses imply that there is a finite upper bound on the diameters of all $\mathbf{G}$s. In the following, $\Delta$ will denote such upper bound. We denote by $\mathcal{C}_{\text{eq.}}(\text{supp}(\mathcal{D}))$ the set of continuous and permutation-equivariant functions over $\text{supp}(\mathcal{D})$.

**Theorem 1.** *Let $\mathcal{D}$ be a distribution over $\mathcal{G}$ for which the above hypotheses hold.*
$$\text{Then if } \overline{\overline{k}} \geq \Delta + 2, \mathcal{H}^{\overline{\overline{k}}} \text{ is dense in } \mathcal{C}_{eq.}(supp(\mathcal{D})).$$

**Sketch of the proof** (see Appendix B.3 for all details) Still following [25], we first prove a modified version of the *Stone-Weierstrass theorem for equivariant functions*. This theorem guarantees that a certain subalgebra of functions is dense in the set of continuous and permutation-equivariant

functions if it separates non-isomorphic Interaction Graphs. Following the idea of [26], we extend the hypothesis space to ensure closure under addition and multiplication. We then prove that the initial hypothesis space is dense in this new subalgebra. Finally, we conclude the proof by showing that the separability property mentioned above is satisfied by this newly-defined subalgebra.

**Corollary 1.** *Let $\mathcal{D}$ be a distribution over $\mathcal{G}$ for which the above hypotheses hold. Let $\ell$ be a continuous and permutation-invariant loss function such that for any $\mathbf{G} \in supp(\mathcal{D})$, problem ([1]) has a unique minimizer $\mathbf{U}^*(\mathbf{G})$, continuous w.r.t $\mathbf{G}$. Then $\forall \epsilon > 0, \exists Solver_\theta \in \mathcal{H}^{\Delta+2}$, such that*

$$\forall \mathbf{G} \in supp(\mathcal{D}), \|Solver_\theta(\mathbf{G}) - \mathbf{U}^*(\mathbf{G})\| \le \epsilon$$

This corollary is an immediate consequence of Theorem [1] and ensures that there exists a DSS using at most $\Delta + 2$ propagation updates that approximates with an arbitrary precision for all $\mathbf{G} \in supp(\mathcal{D})$ the actual solution of problem ([1]). This is particularly relevant when considering large Interaction Graphs that have small diameters.

**Discussion**: This universal approximation theorem does not offer any guarantee of convergence toward the ideal solver $\mathbf{U}^*$ – but there hardly exist such convergence guarantees in the field of Deep Learning. However, this non-trivial result provides a solid theoretical ground to the proposed approach by proving its consistency.

## 5   Experiments

This section investigates the behavior and performances of DSSs on two SSPs. The first one amounts to solving linear systems, though the distribution of problems is generated from a discretized Poisson PDE. The second is the (non-quadratic) AC power flow computation. With respect to the hypotheses of the theoretical results in Section [4], the continuity and the permutation invariance conditions are satisfied in both cases, while the uniqueness can only be proven for the linear system. However it is very likely to hold for many problems.

In all cases, the dataset is split into training/validation/test sets. All free hyperparameters[2] are tuned by trial and errors using the validation set, and **all results presented are results on the test set**. We also compare the DSS to the proxy approach: the architecture is strictly the same, but the loss function used during training is the distance to the "ground truth" (provided by the LU or Newton-Raphson methods). Training is performed using the Adam optimizer [28] with the standard hyperparameters of TensorFlow 1.14 [29], running on an Nvidia GeForce RTX 2080 Ti. Gradient clipping is used to avoid exploding gradient issues. **In the following, all experiments were repeated three times, with the same datasets and different random seeds** (as reported in Tables [1] and [2]). In all experiment and for both the DSS and the proxy approaches, we only report the results of the worst of the three trained models. Our code is in the supplementary materials[3], and links to the datasets are in references.

The main metrics for our experiments are the Pearson correlation and the normalised RMSE (NRMSE) with the output of the classical optimization method (*i.e.* LU in the linear case and Newton-Raphson in the AC Power Flow problems). The NRMSE is computed by dividing the RMSE by the difference of the highest and the lowest values (dividing by the mean for data centered around zero would make no sense). The value of the loss function $\ell$ is computed over the whole test set: the $10^{th}$ and $90^{th}$ percentiles as well as the median are reported.

### 5.1   Solving Linear Systems from a Discretized PDE

**Problem, and goals of experiments** The example SSP considered here comes from the Finite Element Method applied to solve the 2D Poisson equation, one of the simplest and most studied PDE in applied mathematics: the geometry of the domain of the equation is discretized into an unstructured mesh, and computing the vector $\mathbf{U}$ of solution values at each node of the mesh amounts to solving a linear system $\mathbf{AU} = \mathbf{B}$ obtained by assembling local equations [30]. $\mathbf{A}$ and $\mathbf{B}$ encode both the geometry of the problem and the boundary conditions.
For illustration purposes, the Poisson equation can be used to model a field of temperature. In Figure [4], the geometry (house profile) is shown in the Top Left. The result of the optimization is the field of

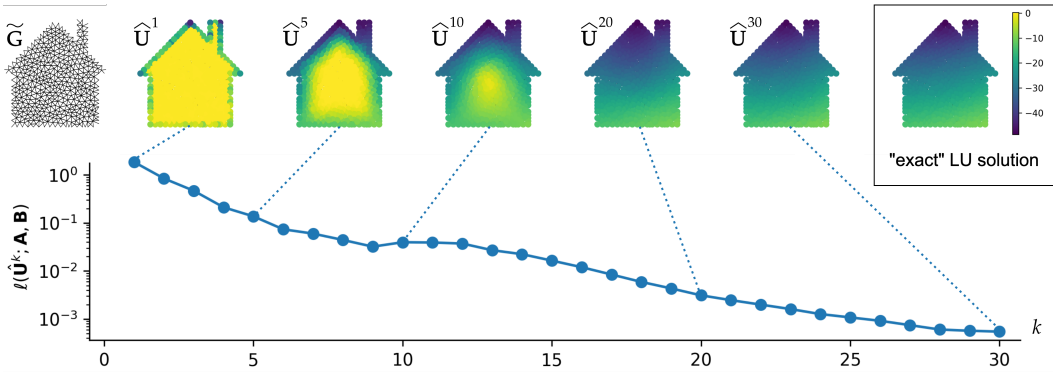

Figure 4: **Intermediate losses and predictions** - Top left: the structure graph $\widetilde{\mathbf{G}}$ (the mesh); Top right: the LU solution; Bottom: evolution of the loss along the $\overline{k} = 30$ updates for a trained DSS, at inference time. The intermediate predictions $\widehat{\mathbf{U}}^k$ are displayed for several values of $k$.

| Method | **DSS** | Proxy | LU | BGS (tol=1e-3) |
|---|---|---|---|---|
| Correlation w/ LU | $>\mathbf{99.99\%}$ | $>99.99\%$ | - | - |
| NRMSE w/ LU | 1.6e-3 | **1.1e-3** | - | - |
| Time per instance (ms) *Inference time divided by batch size | **1.8**$^*$ | **1.8**$^*$ | 2.4 | 2.3 |
| Loss $10^{th}$ percentile | **3.9e-4** | 7.0e-3 | 4.5e-27 | 1.3e-3 |
| Loss $50^{th}$ percentile | **1.2e-3** | 1.6e-2 | 6.1e-26 | 1.7e-2 |
| Loss $90^{th}$ percentile | **4.1e-3** | 4.0e-2 | 6.3e-25 | 1.1e-1 |

Table 1: **Solving specific linear systems** – for similar accuracy, DSS is faster than the iterative BGS thanks to GPU parallelism, while highly correlated with the "exact" solution as given by LU.

temperature everywhere in the house (shown in the Top Right).
This problem is easily set as an SSP in which each node $i$ corresponds to a node of the mesh, all parameters are scalars ($d_A = d_B = d_U = 1$), and the loss function is the following:

$$\ell(\mathbf{U}, \mathbf{G}) = \sum_{i \in [n]} \left( \sum_{j \in [n]} A_{ij} U_j - B_i \right)^2 \tag{10}$$

It is clearly permutation-invariant and satisfies both the unicity of the solution and the continuity conditions evoked in Corollary 1. Our goal here is of course not to solve the Poisson equation, nor is it to propose a new competitive method to invert linear systems. As a matter of fact, the proposed approach does not make use of the linearity of the problem. Our goal is actually twofold: i) validate the DSS approach in high dimension ($n \approx 500$ nodes), and ii) analyze how DSS learns the distribution $\mathcal{D}$. Here, the distribution $\mathcal{D}$ is defined by the specific structure of linear systems that result from the discretization of the Poisson equation. In particular, we will carefully study the generalization capability of the learned model in terms of problem size, for similar problem structures.

**Experimental conditions** The dataset [31] consists of $96180/32060/32060$ training/validation/test examples from the distribution generated from the discretization of the Poisson equation: randomly generated 2D geometries and random values for the second-hand function $f$ and boundary condition $g$ are used to compute the $\mathbf{A}$s and $\mathbf{B}$s. Their number of nodes $n$ are around $500$ (max $599$) (automatic mesh generators do not allow a precise control of $n$).

The number of updates $\overline{k}$ is set to 30 (average diameter size for the considered meshes). Each NN block has one hidden layer of dimension $d = 10$ and a leaky-ReLU non linearity; we have $\alpha = 1\text{e-}3$, lr = 1e-2 and $\gamma = 0.9$. The complete DSS has $49,830$ weights. Training is done for $280,000$ iterations ($48h$) with batch size 100.

Two baseline methods are considered [32], the direct LU decomposition, that could be considered giving the "exact" solution for these sizes of matrices, and the iterative Biconjugate Gradient Stabilized methods (BGS), with stopping tolerances of $10^{-3}$. These algorithms are run on an Intel Xeon Silver 4108 CPU (1.80GHz) (GPU implementations were not available, they could decrease LU computational cost by a factor 6 [33]).

**Results** Table 1 displays comparisons between a trained DSS and the baselines. First, these results validate the approach, demonstrating that DSS can learn to solve 500 dimensional problems rather accurately, and in line with the "exact" solutions as provided by the direct method LU (99.99% correlation). Second, DSS is slighly but consistently faster than the iterative method BGS for similar accuracy (a tunable parameter of BGS). Further work will explore how DSS scales up in much higher

dimensions, in particular when LU becomes intractable. We observe similar results for the proxy approach. Figure 4 illustrates, on a hand-made test example (the mesh is on the upper left corner), how the trained DSS updates its predictions, at inference time, along the $\overline{k}$ updates. The flow of information from the boundary to the center of the geometry is clearly visible.

But what did exactly the DSS learn? Next experiments are concerned with the super-generalization capability of DSSs, looking at their results on test examples sampled from distributions departing from the one used for learning.

**Super-Generalization** We now experimentally analyze how well a trained model is able to generalize to a distribution $\mathcal{D}$ that is different from the training distribution. The same data generation process that was used to generate the training dataset (see above) is now used with meshes of very different sizes, everything else being equal. Whereas the training distribution only contains Interaction Graphs of sizes around 500, out-of-distribution test examples have sizes from 100 and 250 (left of Figure 5) up to 750 and 1000 (right of Figure 5). In all cases, the trained model is able to achieve a correlation with the "true" LU solution as high as 99.99%. Interestingly, the trained DSS achieves a higher correlation with the LU solutions for data points with a lower number of nodes, while the correlation of the

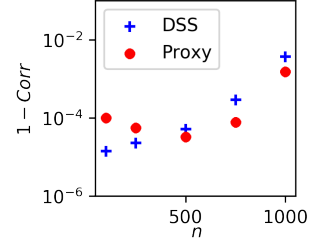

Figure 5: **Varying problem size** $n$: Correlation (DSS, LU)

proxy model decreases when $n$ both increases and decreases. Further experiments with even larger sizes are needed to reach the upper limit of such a super generalization. Nevertheless, thanks to the specific structure dictated to the linear system by the Poisson equation, DSS was able to perform some kind of zero-shot learning for problems of very different sizes.

Other experiments (see Appendix C) were performed by adding noise to $\mathbf{A}$ and $\mathbf{B}$. The performance of the trained model remains good for small noise, then smoothly degrades as the noise increases.

## 5.2 AC power flow experiments

**Problem and goals of experiments** The second SSP example is the AC power flow prediction. The goal is to compute the steady-state electrical flows in a Power Grid, an essential part of real-time operations. Knowing the amount of power that is being produced and consumed throughout the grid (encoded into $\mathbf{B}$, and assumed to be consistent, i.e., production equates consumption), and the way power lines are interconnected, as well as their physical properties (encoded into $\mathbf{A}$), the goal is to compute the voltage defined at each electrical node $V_i = |V_i|e^{\mathbf{j}\theta_i}$ ($\mathbf{j}$ denotes the imaginary unit), which we encode in the states $\mathbf{U}$. Kirchhoff's law (energy conservation at every node) governs this system, and its violation is directly used as loss function $\ell$. Moreover, some constraints over the states $\mathbf{U}$ are here relaxed and included as an additional term of the loss (with factor $\lambda$). One should also keep in mind that the main goal is to predict power flows, and not the voltages per se: Both aspects will be taken into account by measuring the correlation w.r.t $|V_i|$, $\theta_i$, $P_{ij}$ (real part of power flow) and $Q_{ij}$ (imaginary part). This problem is highly non-linear, and a substantial overview is provided in [34]. This set of complex equations can be converted into a SSP using $\mathbf{A}$, $\mathbf{B}$ and $\mathbf{U}$ as defined above ($d_A = 2$, $d_B = 5$, $d_U = 2$), and loss function $\ell$:

$$\ell(\mathbf{U},\mathbf{G}) = \sum_{i \in [n]} (1-B_i^5)\big(-B_i^1 + U_i^1 \sum_{j \in [n]} A_{ij}^1 U_j^1 \cos(U_i^2 - U_j^2 - A_{ij}^2)\big)^2$$
$$+ \sum_{i \in [n]} B_i^3\big(-B_i^2 + U_i^1 \sum_{j \in [n]} A_{ij}^1 U_j^1 \sin(U_i^2 - U_j^2 - A_{ij}^2)\big)^2 + \lambda \sum_{i \in [n]} (1-B_i^3)\big(U_i^1 - B_i^4\big)^2 \quad (11)$$

More details about the conversion from classical power systems notations to this set of variables is provided in Appendix D. This loss is not quadratic, as demonstrated by the presence of sinusoidal terms. One can notice the use of binary variables $B_i^3$ and $B_i^5$. Since both $\mathbf{A}$ and $\mathbf{B}$ vary across the dataset, the problem is largely non-linear with regards to the inputs.

**Experimental conditions** Experiments are conducted on two standard benchmarks from the *Learning to Run a Power Network* competition [35]: *IEEE case14* [36] ($n = 14$), and *IEEE case118* [37] ($n = 118$). In order to increase the diversity in terms of grid topology (encoded in $\mathbf{A}$), for each example, one (resp. two) randomly chosen power lines are disconnected with probability 25% (resp. 25%). For case14 (resp. case118), the dataset is split into 16064/2008/2008 (resp. 18432/2304/2304).

Each NN block has a single hidden layer of dimension $d = 10$ and a leaky-ReLU non linearity. For case14 (resp. case118), $\overline{k}$ was set to 10 (resp. 30) ; we have $\alpha = $ 1e-2, lr = 3e-3 and $\gamma = 0.9$ (resp. $\alpha = $ 3e-4, lr = 3e-3 and $\gamma = 0.9$). The number of weights is $1,722$ for each of the $\overline{k}$ ($\mathbf{M}$, $\mathbf{D}$) blocks,

| Dataset | | IEEE 14 nodes | | | IEEE 118 nodes | | |
|---|---|---|---|---|---|---|---|
| Method | | **DSS** | Proxy | NR | **DSS** | Proxy | NR |
| Corr. w/ NR | $\lvert V_i\rvert$ | 99.93% | >**99.99**% | - | 99.79% | >**99.99**% | - |
| | $\theta_i$ | 99.86% | >**99.99**% | - | 81.31% | >**99.99**% | - |
| | $P_{ij}$ | >**99.99**% | >**99.99**% | - | >**99.99**% | >**99.99**% | - |
| | $Q_{ij}$ | >**99.99**% | >**99.99**% | - | >**99.99**% | >**99.99**% | - |
| NRMSE w/ NR | $\lvert V_i\rvert$ | 2.0e-3 | **4.9e-4** | - | 1.4e-3 | **1.2e-3** | - |
| | $\theta_i$ | 7.1e-3 | **1.7e-3** | - | 5.7e-2 | **4.5e-3** | - |
| | $P_{ij}$ | 6.2e-4 | **2.6e-4** | - | 1.0e-3 | **3.9e-4** | - |
| | $Q_{ij}$ | 4.2e-4 | **2.0e-4** | - | 1.1e-4 | **1.7e-4** | - |
| Time per instance (ms) [*]Inference time divided by batch size | | **1e-2**[*] | **1e-2**[*] | 2e1 | **2e-1**[*] | 2e-1[*] | 2e1 |
| Loss $10^{th}$ percentile | | **4.2e-6** | 2.3e-5 | 1.4e-12 | **1.3e-6** | 6.2e-6 | 2.9e-14 |
| Loss $50^{th}$ percentile | | **1.0e-5** | 4.0e-5 | 2.1e-12 | **1.7e-6** | 8.3e-6 | 4.2e-14 |
| Loss $90^{th}$ percentile | | **4.4e-5** | 1.2e-4 | 3.3e-12 | **2.5e-6** | 1.3e-5 | 6.4e-14 |

Table 2: **Solving specific AC power flow**– our trained DSS models are highly correlated with the Newton-Raphson solutions, while being 2 to 3 orders of magnitude faster thanks to GPU parallelism.

hence $17,220$ (resp. $51,660$) in total. Training is done for $883,000$ (resp. $253000$) iterations with batch size $1,000$ (resp. $500$), and lasted $48h$.

State-of-the-art AC power flow computation uses the Newton-Raphson method, used as baseline here ([38] implementation, on an Intel i5 dual-core (2.3GHz)). To the best of our knowledge, no GPU implementation was available, although recent work [39, 40] investigate such an avenue.

**Results** In both cases, correlations between power flows output by the trained DSSs and the Newton-Raphson method are above $99.99\%$ (both real $P_{ij}$ and imaginary $Q_{ij}$). The same can be said of the proxy models. However, one can observe a less satisfactory correlation in terms of $\lvert V_i\rvert$ and $\theta_i$ for the DSSs while the proxies maintain a correlation higher than $99.99\%$. This can be explained by the fact that the DSSs minimizes power mismatches while the proxies minimize the distance to the Newton-Raphson output in terms of $\lvert V_i\rvert$ and $\theta_i$. However, this does not impact the quality of the power flow prediction, and one should keep in mind that the DSSs learn without any labels, contrarily to the proxies. Table 2 shows the huge acceleration of DSS (by two orders of magnitude) over Newton-Raphson, at the cost of an important decrease in accuracy, although both methods output very similar power flows (correlation higher than $99.99\%$).

## 6 Conclusions and Future Work

This paper proposes a novel paradigm that blends statistics and optimization, Statistical Solver Problems. In the SSP framework, a single solver is trained to solve a family of problem instances sampled from a given distribution of optimization problems, possibly arising from system simulations. Such training is performed by directly minimizing the loss of the optimization problems at hand. In particular, no existing solutions (obtained from costly simulations) are needed for training. The DSSs proposed in this paper, as a particular embodiment of the new proposed framework, is a class of Graph Neural Network, well suited to solving SSPs for which the loss function is permutation-invariant, and for which we theoretically prove some universal approximation properties.

The effectiveness of DSSs are experimentally demonstrated on two problems. Even though experiments on more complex problems are required, the proposed approach shows a good compromise between accuracy and speed in dimensions up to 500 on these two sample problems: solving linear systems, and the non-linear AC power flow. The accuracy on power flow computations matches that of state-of-the-art approaches while speeding up calculations by 2 to 3 orders of magnitude. Our DSS method could also be used as an initialization heuristic for classical optimization algorithms.

Future work will focus on incorporating discrete variables in the state space, and integrating constraints by casting them into a bilevel optimization problem (using two successive DSS that are trained jointly). Other avenues for research concern further theoretical improvements to investigate convergence properties of the DSS approach, in comparison to other solvers, as well as investigations on the limitations of the approach.

## Broader Impact

This work introduces an original approach to solving permutation-invariant problems defined on a graph. The proposed approach is agnostic w.r.t. the practical problem it is applied to. As such, no direct poor societal consequences of this work are to be feared. However, and this is an issue that goes beyond this particular work, it can be applied to critical industrial problems, as demonstrated with the power grid experiments we use to illustrate and validate the approach in Section 5.2; in such context, it is important to ensure by design (*i.e.*, in the definition of the search space and the objective function) that the proper constraints are applied to avoid detrimental solutions. Being able to validate the obtained solution is a problem-specific issue, but validating the whole approach is the holy grail of such work, and is by now out of reach.

## Funding Disclosure

The first author is partly funded by the ANR CIFRE contract 2018/0386.

## Acknowledgement

The authors would like to thank Rémy Clément, Laure Crochepierre, Cédric Josz and the reviewers for their careful reviews and insightful suggestions; and Victor Berger for the fruitful discussions.

## Footnotes

[1]A more rigorous definition of the actual underlying graph structure is deferred to Appendix A

[2]Ranges of tested hyperparameters : $d \in [5, 20]$, $\overline{k} \in [10, 40]$, hidden layers $\in [1, 2]$, $\alpha \in [$1e-1, 1e-4$]$ (see Appendix [E]), non linearity $\in \{$tanh, leaky_ReLU$\}$, $lr \in [$1e-1, 1e-4$]$, $\gamma \in \{0, 0.5, 0.9, 1\}$.

[3]code also available at https://github.com/bdonon/DeepStatisticalSolvers

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
