[Supplementary Material · NeurIPS2020_DeepStatisticalSolver_CameraReady-Appendix.pdf]

# A Underlying Graph Structure

We generalize the standard notion of neighborhood to the setting of Interaction Graphs and SSP $(\mathcal{G}, \mathcal{D}, \mathcal{U}, \ell)$. The intuitive way of defining neighbors of a node $i$ is to look for nodes $j$ such that $A_{ij} \neq 0$ or $A_{ji} \neq 0$. However this intuitive definition does not perfectly suit the case of SSPs as the properties of the loss function $\ell$ do impact the interactions between nodes.

For instance, let the loss function $\ell$ be defined by $\ell(\mathbf{U}, \mathbf{G}) = \sum_{i \in [n]} f(U_i)$, for some real-valued function $f$. In this case, there is no interaction between nodes when computing the state of the Interaction Graph, even though some coefficient $A_{ij}$ may be non zero. We note in this case that:

$$\forall i \neq j, \forall \mathbf{U} \in \mathcal{U}, \frac{\partial^2 \ell}{\partial U_i \partial U_j}(\mathbf{U}, \mathbf{G}) = 0 \tag{12}$$

We thus propose the following definition of the neighborhood of a node $i$ with respect to Interaction Graph $\mathbf{G}$ and loss function $\ell$:

$$\mathcal{N}(i; \mathbf{G}, \ell) = \left\{ j \in [n] \mid \exists \mathbf{U}, \frac{\partial^2 \ell}{\partial U_i \partial U_j}(\mathbf{U}, \mathbf{G}) \neq 0 \right\} \tag{13}$$

For a given class of SSP, the loss function $\ell$ does not change, so it will be omitted in the following, and we will write $\mathcal{N}(i; \mathbf{G})$.

One can observe that in the case of a quadratic optimization problem where $d_A = d_B = d_U = 1$ and $\ell(\mathbf{U}, \mathbf{G}) = \mathbf{U}^T \mathbf{A} \mathbf{U} + \mathbf{B}^T \mathbf{U}$, this notion of neighborhood is exactly that given in Section 2.1, and $\widetilde{\mathbf{G}}$ is indeed the undirected graph defined by the non-zero entries of $\mathbf{A}$ (or more precisely those of $(\mathbf{A} + \mathbf{A}^T)/2$ when $\mathbf{A}$ is not symmetric).

# B Mathematical proofs

In this section, we'll follow Keriven and Peyré [25] and use the notation $[\mathbf{G}]_i$ to denote the $i^{th}$ component of any Interaction Graph or hyper-graph $G$. In the following, 'dense' means 'dense with respect to the uniform metric' by default. As a reminder, the uniform metric $\overline{d}$ on function spaces given two metric spaces $(X, d_X)$ and $(Y, d_Y)$ is defined by

$$\overline{d}(f, f') := \sup_{x \in X} d_Y(f(x), f'(x)). \tag{14}$$

## B.1 Proof of equivariance of the proposed DSS architecture

The following is a proof of the equivariance of the architecture proposed in Section 3.

*Proof.* Because the loss function $\ell$ is permutation invariant, we only have to prove that eq. (8)-(9) satisfy the permutation-equivariance property.

Let us prove by induction on $k$ that $\mathbf{H}^k$ is permutation-equivariant (by a slight abuse of notation in eq. (8), we consider the latent states $\mathbf{H}^k$ as functions of $\mathbf{G}$), *i.e.* that $\mathbf{H}^k(\sigma \star \mathbf{G}) = \sigma \star \mathbf{H}^k(\mathbf{G})$.

For $k = 0$, it is clear that $\sigma \star \mathbf{H}^0 = (0, ..., 0)_{i \in [n]} = \mathbf{H}^0$, which is independant of $\mathbf{G}$.

Now suppose the equivariance property holds for $\mathbf{H}^{k-1}$, then from eq. (5) comes

$$[\phi_{\to}^k(\sigma \star \mathbf{G})]_i = \sum_{j \in \mathcal{N}^\star(i; \sigma \star \mathbf{G})} \Phi_{\to, \theta}^k(H_i^{k-1}(\sigma \star \mathbf{G}), (\sigma \star \mathbf{A})_{ij}, H_j^{k-1}(\sigma \star \mathbf{G})) \tag{15}$$

$$= \sum_{j \in \mathcal{N}^\star(i; \sigma \star \mathbf{G})} \Phi_{\to, \theta}^k([\sigma \star \mathbf{H}^{k-1}(\mathbf{G})]_i, (\sigma \star \mathbf{A})_{ij}, [\sigma \star \mathbf{H}^{k-1}(\mathbf{G})]_j) \tag{16}$$

$$= \sum_{j \in \mathcal{N}^\star(i; \sigma \star \mathbf{G})} \Phi_{\to, \theta}^k(H_{\sigma^{-1}(i)}^{k-1}(\mathbf{G}), A_{\sigma^{-1}(i)\sigma^{-1}(j)}, H_{\sigma^{-1}(j)}^{k-1}(\mathbf{G})) \tag{17}$$

$$= \sum_{\sigma^{-1}(j) \in \mathcal{N}^\star(\sigma^{-1}(i); \mathbf{G})} \Phi_{\to, \theta}^k(H_{\sigma^{-1}(i)}^{k-1}(\mathbf{G}), A_{\sigma^{-1}(i)\sigma^{-1}(j)}, H_{\sigma^{-1}(j)}^{k-1}(\mathbf{G})) \tag{18}$$

$$= \sum_{j \in \mathcal{N}^\star(\sigma^{-1}(i); \mathbf{G})} \Phi_{\to, \theta}^k(H_{\sigma^{-1}(i)}^{k-1}(\mathbf{G}), A_{\sigma^{-1}(i)j}, H_j^{k-1}(\mathbf{G})) \tag{19}$$

$$= [\phi_{\to}^k(\mathbf{G})]_{\sigma^{-1}(i)} \tag{20}$$

$$= [\sigma \star \phi_{\to}^k(\mathbf{G})]_i \tag{21}$$

All the above equalities are straightforward, except maybe eq. (18), which comes from the equivariance property of the notion of neighborhood defined above by eq. (13). The same property follows for $\phi_{\leftarrow}^k$ by similar argument.

For $\phi_{\circlearrowleft}^k$, eq. (7) gives

$$[\phi_{\circlearrowleft}^k(\sigma \star \mathbf{G})]_i = \Phi_{\circlearrowleft, \theta}^k([\mathbf{H}^{k-1}(\sigma \star \mathbf{G})]_i, (\sigma \star \mathbf{A})_{ii}) \tag{22}$$

$$= \Phi_{\circlearrowleft, \theta}^k([\sigma \star \mathbf{H}^{k-1}(\mathbf{G})]_i, (\sigma \star \mathbf{A})_{ii}) \tag{23}$$

$$= \Phi_{\circlearrowleft, \theta}^k(H_{\sigma^{-1}(i)}^{k-1}(\mathbf{G}), A_{\sigma^{-1}(i)\sigma^{-1}(i)}) \tag{24}$$

$$= [\phi_{\circlearrowleft}^k(\mathbf{G})]_{\sigma^{-1}(i)} \tag{25}$$

$$= [\sigma \star \phi_{\circlearrowleft}^k(\mathbf{G})]_{(i)} \tag{26}$$

This concludes the proof that $\mathbf{H}_i^k$ is permutation equivariant for all $k$, and from eq. (8) we conclude that $\mathbf{M}_\theta^k$ is permutation-equivariant. Similar proof holds for $\mathbf{D}_\theta^k$ and $\hat{\mathbf{U}}^k$, which in turn prove that

$$\widehat{\mathbf{U}}(\sigma \star \mathbf{G}) = \sigma \star \widehat{\mathbf{U}}(\mathbf{G}). \tag{27}$$

This concludes the proof. □

## B.2 Proof of Property 1

In Section 3, Property 1 states that if the loss function $\ell$ is permutation-invariant and if for any $\mathbf{G} \in \text{supp}(\mathcal{D})$ there exists a unique minimizer $\mathbf{U}^*(\mathbf{G})$ of problem (1), then $\mathbf{U}^*$ is permutation-equivariant.

Let $\ell$ be a permutation-invariant loss function and $\mathcal{D}$ a distribution such that for any $\mathbf{G} \in \text{supp}(\mathcal{D})$ there is a unique solution $\mathbf{U}^*$ of problem 1. Let $\mathbf{G} = (n, \mathbf{A}, \mathbf{B}) \in \text{supp}(\mathcal{D})$ and $\sigma \in \Sigma_n$ a permutation.

$$\ell(\sigma \star \mathbf{U}^*(\mathbf{G}), \sigma \star \mathbf{G}) = \ell(\mathbf{U}^*(\mathbf{G}), \mathbf{G}) \qquad \text{by invariance of } \ell \tag{28}$$

$$= \min_{\mathbf{U} \in \mathcal{U}} \ell(\mathbf{U}, \mathbf{G}) \qquad \text{by definition of } \mathbf{U}^* \tag{29}$$

$$= \min_{\mathbf{U} \in \mathcal{U}} \ell(\sigma \star \mathbf{U}, \sigma \star \mathbf{G}) \qquad \text{by invariance of } \ell \tag{30}$$

$$= \min_{\mathbf{U} \in \mathcal{U}} \ell(\mathbf{U}, \sigma \star \mathbf{G}) \qquad \text{by invariance of } \mathcal{U} \tag{31}$$

$$= \ell(\mathbf{U}^*(\sigma \star \mathbf{G}), \sigma \star \mathbf{G}) \qquad \text{by definition of } \mathbf{U}^* \tag{32}$$

Moreover the uniqueness of the solution ensures that $\mathbf{U}^*(\sigma \star \mathbf{G}) = \sigma \star \mathbf{U}^*(\mathbf{G})$, which concludes the proof.

## B.3 Proof of Theorem 1

We will now prove Theorem 1, the main result on DSSs, by closely following the approach of [25]. We will first prove a modified version of the Stone-Weierstrass theorem, and then verify that the defining spaces for Interaction Graphs indeed verify the conditions of this theorem by proving several lemmas (most importantly Theorem 3 on separability).

Let $\mathcal{G}_{\text{eq.}} \subseteq \mathcal{G}$ be a set of compact, permutation-invariant Interaction Graphs. The compactness implies that there exist $\overline{n} \in \mathbb{N}$ such that all graphs in $\mathcal{G}_{\text{eq.}}$ have an amount of nodes lower than $\overline{n} \in \mathbb{N}$. Let $\mathcal{C}_{\text{eq.}}(\mathcal{G}_{\text{eq.}}, \mathcal{U})$ be the space of continuous functions from $\mathcal{G}_{\text{eq.}}$ on $\mathcal{U}$ that associate to any Interaction Graph $\mathbf{G} = (n, \mathbf{A}, \mathbf{B})$ one of its possible states $\mathbf{U} \in \mathbb{R}^n$. $(\mathcal{C}_{\text{eq.}}(\mathcal{G}_{\text{eq.}}, \mathcal{U}), +, \cdot, \odot)$ is a unital $\mathbb{R}$-algebra, where $(+, \cdot)$ are the usual addition and multiplication by a scalar, and $\odot$ is the Hadamard product defined by $[(f \odot g)(x)]_i = [f(x)]_i \cdot [g(x)]_i$. Its unit is the constant function $\mathbf{1} = (1, \ldots, 1)$.

**Theorem 2** (Modified Stone-Weierstrass theorem for equivariant functions).
*Let $\mathcal{A}$ be a unital subalgebra of $\mathcal{C}_{eq.}(\mathcal{G}_{eq.}, \mathcal{U})$, (i.e., it contains the unit function $\mathbf{1}$) and assume both following properties hold:*

- *(Separability) For all $\mathbf{G}, \mathbf{G}' \in \mathcal{G}_{eq.}$, with number of nodes $n$ and $n'$ such that $\mathbf{G}$ is not isomorphic to $\mathbf{G}'$, and for all $k \in [n], k' \in [n']$, there exists $f \in \mathcal{A}$ such that $[f(\mathbf{G})]_k \neq [f(\mathbf{G}')]_{k'}$;*

- *(Self-separability) For all $n \leq \overline{n}$, $I \subseteq [n]$, $\mathbf{G} \in \mathcal{G}_{eq.}$ with $n$ nodes, such that no isomorphism of $\mathbf{G}$ exchanges at least one index between $I$ and $I^c$, and for all $k \in I$, $l \in I^c$, there exists $f \in \mathcal{A}$ such that $[f(\mathbf{G})]_k \neq [f(\mathbf{G})]_l$.*

*Then $\mathcal{A}$ is dense in $\mathcal{C}_{eq.}(\mathcal{G}_{eq.}, \mathcal{U})$ with respect to the uniform metric.*

This proof of Theorem 2 is almost identical to that of Theorem 4 in [25], with the following differences.

1. For the input space, we consider Interaction Graphs of the form $(n, \mathbf{A}, \mathbf{B})$ with $\mathbf{A} \in (\mathbb{R}^{d_A})^{n^2}$ and $\mathbf{B} \in (\mathbb{R}^{d_B})^n$, instead of hyper-graphs of the form $\mathbb{R}^{n^d}$ for $d \in \mathbb{N}$. The corresponding metrics are naturally different, although the difference is not critical for the proof;

2. Similarly, we consider an output space with $\mathbf{U} \in (\mathbb{R}^{d_U})^n$ instead of $\mathbb{R}^n$;

3. We only assume $\mathcal{G}_{\text{eq.}} \subseteq \mathcal{G}$ to be compact and permutation-invariant instead of a $\mathcal{G}_{\text{eq.}}$ with an explicit form: $\mathcal{G}_{\text{eq.}} := \{G \in \mathbb{R}^{n^d} | n \leq n_{\max}, \|G\| \leq R\}$ (which makes this modified theorem more general).

We shall then indicate how to bypass these differences one by one and then reuse the proofs in [25].

For 1, the only properties of the input space involved in [25] are the number of nodes, action of permutation and the metric (with the corresponding topology). For the first two points, everything is still applicable in our setting. For the topology, the difference is not critical either since we are actually considering the product space of two of metric spaces defined in [25] and all corresponding properties follow.

For 2, we can always reduce to the case with $d_U = 1$ then stack the resulting function $d_U$ times to have the expected shape. This works seamlessly with Hadamard product and all properties related to density.

For 3, there is actually no dependency on the explicit form of $\mathcal{G}_{\text{eq.}}$ or $\mathbf{G}$ in [25] (as for the case in 1). And the proof only relies on the upper bound on the number of nodes. So this generalization can be naturally obtained.

The detailed proof of Theorem 2 then follows the exact same procedure than that of Theorem 4 in [25], and we shall omit it here, refering the reader to [25] for all details.

Let $\overline{\overline{k}} \in \mathbb{N}$, and, as defined in Section 4, let $\mathcal{H}^{\overline{\overline{k}}}$ be the set of graph neural networks defined in Section 3 such that $\overline{k} \leq \overline{\overline{k}}$. Our goal is to prove that Theorem 2 can be applied to $\mathcal{H}^{\overline{\overline{k}}}$.

Because $\mathcal{H}^{\overline{\overline{k}}}$ is not an algebra, let us consider $\mathcal{H}^{\overline{\overline{k}}\odot}$, the algebra generated by $\mathcal{H}^{\overline{\overline{k}}}$ with respect to the Hadamard product. More formally:

$$\mathcal{H}^{\overline{\overline{k}}\odot} = \left\{ \sum_{s=1}^{S} \bigodot_{t=1}^{T_s} c_{st} f_{st} \,|\, S \in \mathbb{N}, T_s \in \mathbb{N}, c_{st} \in \mathbb{R}, f_{st} \in \mathcal{H}^{\overline{\overline{k}}} \right\}. \tag{33}$$

Note that the Hadamard product among $f_{st}$'s is well-defined since for a fixed input $\mathbf{G}$, all output values $f_{st}(\mathbf{G})$ take the same dimension - the size of $\mathbf{G}$.

$(\mathcal{H}^{\overline{\overline{k}}\odot}, +, \cdot, \odot)$ is obviously a unital sub-algebra of $(\mathcal{G}_{\text{eq.}}, +, \cdot, \odot)$ (the constant function $(1, \ldots, 1)$ trivially belongs to $\mathcal{H}^{\overline{\overline{k}}\odot}$). In order to apply Theorem 2 to $\mathcal{H}^{\overline{\overline{k}}\odot}$, one needs to prove that it satisfies both separability hypotheses.

Let us first notice that the self-separability property is a straightforward consequence of the hypothesis of separability of external inputs on $\text{supp}(\mathcal{D})$. Hence we only need to prove the separability property:

**Theorem 3.** $\mathcal{H}^{\overline{\overline{k}}\odot}$ *satisfies the separability property of Theorem 2.*

The proof consists of 3 steps. In step 1, we prove that for all $\mathbf{G}, \mathbf{G}' \in \text{supp}(\mathcal{D})$ that are not isomorphic, there exists a sequence *(node, edge, node)* that only exists in $\mathbf{G}$. In Step 2, we build a continuous function $f^{\dagger}$ on $\mathcal{G}$ that returns an indicator of the presence of this sequence in the input graph. In Step 3, we prove that there exists a function $f_{\theta} \in \mathcal{H}^{\overline{\overline{k}}\odot}$ that approximates well enough $f^{\dagger}$.

For Step 1, we formally state it in the following lemma.

**Lemma 1.** *Let $\mathbf{G} = (n, \mathbf{A}, \mathbf{B})$ and $\mathbf{G}' = (n', \mathbf{A}', \mathbf{B}')$ be in $\text{supp}(\mathcal{D})$ such that $\mathbf{G}$ and $\mathbf{G}'$ are not isomorphic and $n \geq n'$. Then there exist $i, j \in [n]$, $i \neq j$, such that, for all $i', j' \in [n']$, the following inequality holds:*

$$(B_i, A_{ij}, B_j) \neq (B'_{i'}, A'_{i'j'}, B'_{j'}) \tag{34}$$

*Proof.* This lemma relies on the separability hypothesis of $\text{supp}(\mathcal{D})$ which states that there exists $\delta > 0$ such that for all $\mathbf{G} = (n, \mathbf{A}, \mathbf{B}) \in \text{supp}(\mathcal{D})$ and for all $i \neq j \in [n]$, $\|B_i - B_j\| \geq \delta$.

We shall use proof by contradiction: assume that for any $(i, j) \in [n]^2$ with $i \neq j$, there exists $\alpha(i, j) = (i', j') \in [n']^2$ such that $(B_i, A_{ij}, B_j) = (B'_{i'}, A'_{i'j'}, B'_{j'})$. Two cases must be distinguished, depending on whether $n < n'$ or $n = n'$

If $n > n'$, then according to the pigeonhole principle, there exist two pairs $(i, j) \in [n]^2$ and $(l, m) \in [n]^2$ that have the same image by $\alpha$, $(i', j') \in [n']^2$. Hence, $(B_i, A_{ij}, B_j) = (B'_{i'}, A'_{i'j'}, B'_{j'}) = (B_l, A_{lm}, B_m)$, which contradicts the separability hypothesis for $\mathbf{G}$.

If $n = n'$, according to the separability hypothesis of $\text{supp}(\mathcal{D})$, there cannot exist $i \neq l \in [n]$ that are mapped to the same $i' \in [n']$ (*i.e.* $\alpha(i, j) = (i', j')$ for some $j, j'$ and $\alpha(l, m) = (i', m')$ for some $m, m'$). Thus $\alpha$ actually defines an injective mapping $\chi : [n] \to [n]$ on the first component. Because $n = n'$, this mapping is also surjective and hence bijective. Due to the symmetry of $i$ and $j$, we see that the mapping on the second component $\chi'$ defined by $X$ is exactly $\chi$. Hence we have found a permutation $\chi \in \Sigma_n$ such that

$$B_i = B'_{\chi(i)} \tag{35}$$

$$A_{ij} = A_{\chi(i)\chi(j)} \tag{36}$$

for any $(i, j) \in [n]^2$, which means that $\mathbf{G}$ and $\mathbf{G}'$ are isomorphic, contradicting the hypothesis, and thus completing the proof. $\square$

Let us now proceed with Step 2. For convenience, we shall use a continuous kernel function defined by

$$K_{\epsilon}(x) = \max(0, 1 - |x|/\epsilon) \tag{37}$$

for $\epsilon > 0$. Then we have $K_{\epsilon}(0) = 1$ and $K_{\epsilon}(x) = 0$ for $|x| > \epsilon$.

All intermediate functions of DSSs $\Phi_{\rightarrow}^k, \Phi_{\leftarrow}^k, \Phi_{\circlearrowleft}^k, \Psi^k$ and $\Xi^k$ (Section 3) live in function spaces that satisfy the Universal Approximation Property (UAP). So let us consider now a space of continuous

Figure 6: **Kernel function**

functions that share the same architecture than DSS, but in which all spaces of parameterized neural networks have been replaced by corresponding continuous function space. We denote this space by $\mathcal{H}^{\overline{\overline{k}}\dagger}$ (by convention, a dagger($\dagger$) added to a Neural Network block from Section 3 will refer to the corresponding continuous function space (*e.g.* $\Phi_{\rightarrow}^{k\dagger}$). We are now in position to prove the following lemma.

**Lemma 2.** *For any* $\mathbf{G}, \mathbf{G}' \in supp(\mathcal{D})$ *that are not isomorphic, there exists a function* $f^{\dagger} \in \mathcal{H}^{\overline{\overline{k}}\dagger}$ *such that for any* $k \in [n], k' \in [n']$*, we have* $[f^{\dagger}(\mathbf{G})]_k \neq [f^{\dagger}(\mathbf{G}')]_{k'}$.

*Proof.* Without loss of generality, we suppose $n \geq n'$. According to Lemma 1, there exist $(i^{\dagger}, j^{\dagger}) \in [n]^2$, $i^{\dagger} \neq j^{\dagger}$, such that $\mathbf{G}$ contains a sequence $(B_{i^{\dagger}}, A_{i^{\dagger}j^{\dagger}}, B_{j^{\dagger}})$ that does not appear in $\mathbf{G}'$.

We are going to construct a continuous function $f^{\dagger} : supp(\mathcal{D}) \to \mathcal{U}$ that will be an indicator of the presence of the above sequence in the graph, and such that $f^{\dagger}(\mathbf{G}) = (1, \ldots, 1) \in \mathbb{R}^n$ and $f^{\dagger}(\mathbf{G}') = (0, \ldots, 0) \in \mathbb{R}^{n'}$ (thus proving Lemma 2).

Let us first recall the architecture of DSS, as defined by eq. (5)-(9), and let us choose *continuous* functions $\Phi_{\rightarrow}^{1\dagger}, \Phi_{\leftarrow}^{1\dagger}, \Phi_{\circlearrowleft}^{1\dagger}$ and $\Psi^{1\dagger}$ such that $\mathbf{H}^{1\dagger}$ is defined by, for any $\mathbf{G}'' \in supp(\mathcal{D})$,

$$[\mathbf{H}^{1\dagger}(\mathbf{G}'')]_i = 2K_{\epsilon}(\|B_i'' - B_{i^{\dagger}}\|) - K_{\epsilon}(\|B_i'' - B_{j^{\dagger}}\|) \tag{38}$$

where $\epsilon = \|A_{i^{\dagger}j^{\dagger}} - A'_{\sigma(i^{\dagger})\sigma(j^{\dagger})}\|$ if $\mathbf{B}$ and $\mathbf{B}'$ are isomorphic through permutation $\sigma$ and $\epsilon = \min_{\sigma} \max_i \|B_i - B'_{\sigma(i)}\|$ otherwise. This function allows us to identify whether the external input $B_i''$ is close to one of $B_{i^{\dagger}}$ or $B_{j^{\dagger}}$.

For $k = 2$, we define

$$\Phi_{\rightarrow}^{2\dagger}(h, a, h') = K_{\epsilon}(\|h - 2\| + \|a - A_{i^{\dagger}j^{\dagger}}\| + \|h' + 1\|) \tag{39}$$

and

$$\Phi_{\circlearrowleft}^{2\dagger}(h, a) = K_{\epsilon}(\|h - 1\| + \|a - A_{i^{\dagger}j^{\dagger}}\|). \tag{40}$$

Then we choose $\Psi^{2\dagger}$ such that

$$[\mathbf{H}^{2\dagger}]_i = \phi_{\circlearrowleft,i}^{2\dagger} + \phi_{\rightarrow,i}^{2\dagger} = \Phi_{\circlearrowleft}^{2\dagger}(H_i^{1\dagger}, A_{ii}) + \sum_{j \in \mathcal{N}^{\star}(i;\mathbf{G})} \Phi_{\rightarrow}^{2\dagger}(H_i^{1\dagger}, A_{ij}, H_j^{1\dagger}) \tag{41}$$

According to the construction of $\mathbf{H}^{1\dagger}$ and $\mathbf{H}^{2\dagger}$, we have $[\mathbf{H}^{2\dagger}(G)]_i = 1$ if $i = i^{\dagger}$ and 0 otherwise. And $\mathbf{H}^{2\dagger}(G') = (0, \ldots, 0)$.

For $k \geq 3$, we let

$$[\mathbf{H}^{k+1\dagger}]_i = [\mathbf{H}^{k\dagger}]_i + \sum_{j \in \mathcal{N}^{\star}(i;\mathbf{G})} [\mathbf{H}^{k\dagger}]_j \tag{42}$$

Thus if $\overline{\overline{k}} \geq \Delta + 2$, we have $[\mathbf{H}^{\overline{\overline{k}}\dagger}(\mathbf{G})]_i \geq 1$ for any $i \in [n]$, due to the connectivity and the fact that the diameter of $\mathbf{G}$ is bounded by $\Delta$, *i.e.* the propagation process described in eq. (42) reaches every node of $\mathbf{G}$. We have $[\mathbf{H}^{\overline{\overline{k}}\dagger}(G)]_i \geq 1$ for any $i \in [n]$, and $[\mathbf{H}^{\overline{\overline{k}}\dagger}(G')]_i = 0$ for any $i \in [n']$

Finally for the decoder, we let

$$\Xi^{\overline{\overline{k}}\dagger}(h) = \min(1, h) \tag{43}$$

and

$$[\hat{\mathbf{U}}^{\overline{\overline{k}}\dagger}]_i = \Xi^{\overline{\overline{k}}\dagger}(H_i^{\overline{\overline{k}}\dagger}). \tag{44}$$

We have thus constructed a function $f^{\dagger}$ such that $f^{\dagger}(\mathbf{G}) = (1, \ldots, 1) \in \mathbb{R}^n$ and $f^{\dagger}(\mathbf{G}') = (0, \ldots, 0) \in \mathbb{R}^{n'}$. Thus for any $k \in [n], k' \in [n']$, we have $[f^{\dagger}(\mathbf{G})]_k = 1 \neq 0 = [f^{\dagger}(\mathbf{G}')]_{k'}$, which concludes the proof. $\square$

**Lemma 3.** *Let $X, Y, Z$ be three metric spaces. Let $\mathcal{F} \subseteq \mathcal{C}(X, Y)$ and $\mathcal{G} \subseteq \mathcal{C}(Y, Z)$ be two sets of continuous functions. And let $\mathcal{F}^{\ell} \subseteq \mathcal{F}, \mathcal{G}^{\ell} \subseteq \mathcal{G}$ be two subsets of Lipschitz functions that are dense in $\mathcal{F}$ and $\mathcal{G}$ respectively. Then $\mathcal{G}^{\ell} \circ \mathcal{F}^{\ell} := \{g \circ f | g \in \mathcal{G}^{\ell}, f \in \mathcal{F}^{\ell}\}$ is dense in $\mathcal{G} \circ \mathcal{F}$.*

*Proof.* Let $g \circ f$ be a continuous function in $\mathcal{G} \circ \mathcal{F}$, $\epsilon > 0$. Due to the density of $\mathcal{G}^\ell$ in $\mathcal{G}$, there exists $g^\ell \in \mathcal{G}^\ell$ such that

$$\overline{d}(g, g^\ell) < \frac{\epsilon}{2}. \tag{45}$$

Let $L_{g^\ell}$ be the Lipschitz constant of $g^\ell$, the density of $\mathcal{F}^\ell$ in $\mathcal{F}$ implies that there exists $f^\ell$ such that

$$\overline{d}(f, f^\ell) < \frac{\epsilon}{2L_{g^\ell}}. \tag{46}$$

Then we have

$$d_Z(g \circ f(x), g^\ell \circ f^\ell(x)) \leq d_Z(g \circ f(x), g^\ell \circ f(x)) + d_Z(g^\ell \circ f(x), g^\ell \circ f^\ell(x)) \tag{47}$$

$$< \frac{\epsilon}{2} + L_{g^\ell} d_Y(f(x), f^\ell(x)) \tag{48}$$

$$< \frac{\epsilon}{2} + L_{g^\ell} \frac{\epsilon}{2L_{g^\ell}} = \epsilon \tag{49}$$

for any $x \in X$. Thus $\overline{d}(g \circ f, g^\ell \circ f^\ell) < \epsilon$. Hence $\mathcal{G}^\ell \circ \mathcal{F}^\ell$ is dense in $\mathcal{G} \circ \mathcal{F}$.

$\square$

**Lemma 4.** $\mathcal{H}^{\overline{\overline{k}}}$ *is dense in* $\mathcal{H}^{\overline{\overline{k}}\dagger}$.

*Proof.* As functions in $\mathcal{H}^{\overline{\overline{k}}}$ are composition of Lipschitz functions (neural network with linear transformation and Lipschitz activation as assumed), and all intermediate function spaces verify the Universal Approximation Property. We conclude immediately from using the definition of $\mathcal{H}^{\overline{\overline{k}}\dagger}$ and applying Lemma 3 consecutively. $\square$

We are ready to prove Theorem 3, i.e., that $\mathcal{H}^{\overline{\overline{k}}\odot}$ satisfies the separability hypothesis of Theorem 2.

*Proof.* of Theorem 3

It suffices to show the separability for $\mathcal{H}^{\overline{\overline{k}}}$ since it is a subset of $\mathcal{H}^{\overline{\overline{k}}\odot}$.

Let $\mathbf{G}, \mathbf{G}' \in \text{supp}(\mathcal{D})$. According to Lemma 2, there exists $f^\dagger \in \mathcal{H}^{\overline{\overline{k}}\dagger}$ such that for any $k \in [n], k' \in [n']$, we have $[f^\dagger(\mathbf{G})]_k \neq [f^\dagger(\mathbf{G}')]_{k'}$. According to Lemma 4, there exists $f \in \mathcal{H}^{\overline{\overline{k}}}$ such that

$$\overline{d}(f^\dagger, f) < \frac{1}{3}. \tag{50}$$

Then for any $k \in [n], k' \in [n']$, we have $[f(\mathbf{G})]_k > \frac{2}{3}$ and $[f(\mathbf{G}')]_{k'} < \frac{1}{3}$. This proves the separability of $\mathcal{H}^{\overline{\overline{k}}}$ and furthermore, $\mathcal{H}^{\overline{\overline{k}}\odot}$. $\square$

Before being able to prove Theorem 1, we need the last following lemma.

**Lemma 5.** $\mathcal{H}^{\overline{\overline{k}}}$ *is dense in* $\mathcal{H}^{\overline{\overline{k}}\odot}$.

*Proof.* We shall prove this result by explicitly constructing an approximation function in $\mathcal{H}^{\overline{\overline{k}}}$ for a given function in $\mathcal{H}^{\overline{\overline{k}}\odot}$.

Let $f^\odot \in \mathcal{H}^{\overline{\overline{k}}\odot}$, and $\epsilon > 0$. By definition of $\mathcal{H}^{\overline{\overline{k}}\odot}$ in eq. (33), there exists $S \in \mathbb{N}$, $\{T_s\}_{s \in \{1,\dots,S\}} \in \mathbb{N}^S$, as well as $\{c_{st}\} \in \mathbb{R}$ and $\{f_{st}\} \in \mathcal{H}^{\overline{\overline{k}}}$ for all $(s, t)$ with $s \in [S], t \in [T_s]$, such that :

$$f^\odot = \sum_{s=1}^{S} \bigodot_{t=1}^{T_s} c_{st} f_{st} \tag{51}$$

Thus, for any $(s, t)$, there exists $\overline{k}_{st} \leq \overline{\overline{k}}$, and $d_{st} \in \mathbb{N}$, such that $f_{st}$ is composed of functions $\{\Phi_{\rightarrow,\theta}^{k,s,t}, \Phi_{\leftarrow,\theta}^{k,s,t}, \Phi_{\circlearrowright,\theta}^{k,s,t}, \Psi_{\theta}^{k,s,t}, \Xi_{\theta}^{k,s,t}\}_{k \in [\overline{k}_{st}]}$, as defined by eq. (5)-(9) and Figure 3 in Section 3. $d_{st}$ is the dimension of the latent states of *channel* $f_{st}$.

The different channels can have different number of propagation updates $\bar{k}_{st}$, but they are all bounded by $\bar{\bar{k}}$. Without loss of generality, we can assume that all $\bar{k}_{st}$ are equal to $\bar{\bar{k}}$ by padding, when needed, exactly $\bar{\bar{k}} - \bar{k}_{st}$ null operations $\Phi^k_{\rightarrow}$, $\Phi^k_{\leftarrow}$ and $\Psi^k$ before the actual ones.

Let $d = \sum_{s=1}^{S} \sum_{t=1}^{T_s} d_{st}$ be the cumulated dimensions of the different channels.

For each $(s,t)$, we introduce the matrix $W_{st} \in \{0,1\}^{d_{st} \times d}$ which is defined by:

$$[W_{st}]_{ij} = \begin{cases} 1, & \text{if } \sum_{s'=1}^{s} \sum_{t'=1}^{T_{s'}} d_{s't'} + \sum_{t'=1}^{t-1} d_{st'} + i = j \\ 0, & \text{otherwise.} \end{cases} \tag{52}$$

Thus $W_{st} = [0, \ldots, 0, I_{d_{st}}, 0, \ldots, 0]$. Basically, when given a vector of dimension $d$, $W_{st}$ will be able to select exactly the component that corresponds to the channel $(s,t)$, and will thus return a vector of dimension $d_{st}$.

Let us now define the functions $\{\Phi^k_{\rightarrow,\theta}, \Phi^k_{\leftarrow,\theta}, \Phi^k_{\circlearrowleft,\theta}, \Psi^k_\theta, \Xi^k_\theta\}_{k \in [\bar{\bar{k}}]}$ such that

$$\Phi^k_{\rightarrow,\theta}(H_i^{k-1}, A_{ij}, H_j^{k-1}) = \sum_{s=1}^{S} \sum_{t=1}^{T_s} W_{st}^\top . \Phi^{k,s,t}_{\rightarrow,\theta}(W_{st}.H_i^{k-1}, A_{ij}, W_{st}.H_j^{k-1}) \tag{53}$$

$$\Phi^k_{\leftarrow,\theta}(H_i^{k-1}, A_{ij}, H_j^{k-1}) = \sum_{s=1}^{S} \sum_{t=1}^{T_s} W_{st}^\top . \Phi^{k,s,t}_{\leftarrow,\theta}(W_{st}.H_i^{k-1}, A_{ij}, W_{st}.H_j^{k-1}) \tag{54}$$

$$\Phi^k_{\circlearrowleft,\theta}(H_i^{k-1}, A_{ij}) = \sum_{s=1}^{S} \sum_{t=1}^{T_s} W_{st}^\top . \Phi^{k,s,t}_{\circlearrowleft,\theta}(W_{st}.H_i^{k-1}, A_{ij}) \tag{55}$$

$$\Psi^k_\theta(H_i^{k-1}, B_i, \phi^k_{\rightarrow,i}, \phi^k_{\leftarrow,i}, \phi^k_{\circlearrowleft,i}) = \sum_{s=1}^{S} \sum_{t=1}^{T_s} W_{st}^\top . \Psi^{k,s,t}_\theta(W_{st}.H_i^{k-1}, B_i, W_{st}.\phi^k_{\rightarrow,i}, W_{st}.\phi^k_{\leftarrow,i}, W_{st}.\phi^k_{\circlearrowleft,i}) \tag{56}$$

These functions, using eq. (5)-(8), define a function acting on a latent space of dimension $d$. Moreover, for any channel $(s,t)$ and any node $i \in [n]$, we have $W_{st}.H_i^{\bar{\bar{k}}} = H_i^{\bar{\bar{k}},s,t}$.

We have thus built a function of $\mathcal{H}^{\bar{\bar{k}}}$ that exactly replicates the steps performed on the different channels. Now, let us take a closer look at the decoding step.

Observing that the mapping from $\mathbb{R}^d$ to $\mathbb{R}^{d_U}$, $h \mapsto \sum_{s=1}^{S} \bigodot_{t=1}^{T_s} c_{st} \Xi^{\bar{\bar{k}},s,t}_\theta(W_{st}h)$ is indeed continuous, there exists a mapping $\Xi^{\bar{\bar{k}}}_\theta \in \mathcal{H}^{d_U}_d$ such that :

$$\left\| \Xi^{\bar{\bar{k}}}_\theta(h) - \sum_{s=1}^{S} \bigodot_{t=1}^{T_s} c_{st} \Xi^{\bar{\bar{k}},s,t}_\theta(W_{st}h) \right\| \leq \epsilon \tag{57}$$

for any $h$ in a compact of $\mathbb{R}^d$. The resulting function $f \in \mathcal{H}^{\bar{\bar{k}}}$, composed of $\{\Phi^k_{\rightarrow,\theta}, \Phi^k_{\leftarrow,\theta}, \Phi^k_{\circlearrowleft,\theta}, \Psi^k_\theta, \Xi^k_\theta\}_{k \in [\bar{\bar{k}}]}$ using eq. (5)-(9), approximates $f^\odot$ with precision less than $\epsilon$, which concludes the proof.

$\square$

We now have all necessary ingredients to prove Theorem 1.

*Proof.* According to the hypotheses of compactness and permutation-invariance on $\text{supp}(\mathcal{D})$, both conditions of Theorem 2 are satisfied by $\text{supp}(\mathcal{D})$. Consider the subalgebra $\mathcal{H}^{\bar{\bar{k}}\odot}$ defined by eq. (33). According to the hypothesis of separability of external inputs, the hypothesis of connectivity and Theorem 3, $\mathcal{H}^{\bar{\bar{k}}\odot}$ satisfies the separability and self-separability conditions of Theorem 2. Applying Theorem 2, it comes that $\mathcal{H}^{\bar{\bar{k}}\odot}$ is dense in $\mathcal{C}_{\text{eq.}}(\text{supp}(\mathcal{D}))$. Then according to Lemma 5, $\mathcal{H}^{\bar{\bar{k}}}$ is dense in $\mathcal{H}^{\bar{\bar{k}}\odot}$. We conclude that $\mathcal{H}^{\bar{\bar{k}}}$ is dense in $\mathcal{C}_{\text{eq.}}(\text{supp}(\mathcal{D}))$ by the transitivity property of density. $\square$

### B.4 Proof of Corollary 1

*Proof.* Let $\epsilon > 0$. From Property 1, $\mathbf{U}^*$ is permutation-equivariant. Moreover, by hypothesis, $\mathbf{U}^*$ is continuous. Thus $\mathbf{U}^* \in \mathcal{C}_{\text{eq.}}(\text{supp}(\mathcal{D}))$.

And from Theorem 1, we know that there exists a function $Solver_\theta \in \mathcal{H}^{\Delta+2}$ such that

$$\forall \mathbf{G} \in \text{supp}(\mathcal{D}), \|Solver_\theta(\mathbf{G}) - \mathbf{U}^*(\mathbf{G})\| \leq \epsilon \tag{58}$$

$\square$

## C  Linear Systems derived from the Poisson Equation

This appendix details the experiments of Section 5.1: it presents the data generation process, and also explains the change of variables that was made to help normalizing the data (not mentioned in the main paper for space reason, as it does not change the overall conclusions of the experiments). Finally, we also discuss an additional super generalization experiment briefly cited in the paper.

### C.1  Data generation

**Initial problem**   Consider a Poisson's equation with Dirichlet condition on its boundary $\partial\Omega$:

$$-\triangle u = f \ in \ \Omega$$

$$u|_{\partial\Omega} = g$$

where $\Omega$ a spatial domain in $\mathbb{R}^2$, and $\partial\Omega$ its boundaries. The right hand side $f$ is defined on $\Omega$, and the Dirichlet boundary condition $g$ is defined on $\partial\Omega$. $x$ and $y$ will denote the classical 2D coordinates.

**Random geometries**   Random 2D domains $\Omega$ are generated from 10 points, randomly sampled in the unit square. The Bézier curve that passes through these pints is created, and is further subsampled to obtain approximately 100 points in the unit square. These points defines a polygon, that is used as the boundary $\partial\Omega$. See the left part of Figure 7 to see four instances.

**Random $f$ and $g$**   Functions $f$ and $g$ are defined by the following equations:

$$f(x,y) = r_1(x-1)^2 + r_2 y^2 + r_3, \qquad\qquad (x,y) \in \Omega \tag{59}$$
$$g(x,y) = r_4 x^2 + r_5 y^2 + r_6 xy + r_7 x + r_8 y + r_9, \qquad (x,y) \in \partial\Omega \tag{60}$$

in which parameters $r_i$ are uniformly sampled between -10 and 10.

**Discretization**   The random 2D geometries are discretized using Fenics' standard mesh generation method (see Figure 7-right).

Figure 7: **Discretization of randomly generated domains**

**Assembling** The assembling step [30] consists in building a linear system from the partial differentiate equation and the discretized domain. The unknown are the values of the solution at the nodes of the mesh, and the equations are obtained by using the variational formulation of the PDE on basis functions with support in the neighbors of each node. This is also automatically performed using Fenics. The result of the assembling step is a square matrix $\mathbf{A}$ and a vector $\mathbf{B}$, and the solution is the vector $\mathbf{U}$ such that $\mathbf{A}\mathbf{U} = \mathbf{B}$. Thus, as stated in Section 5, in the framework of SSPs, an Interaction Graph is defined from the number of nodes of the mesh, the matrix $\mathbf{A}$ and the vector $\mathbf{B}$, and the loss function is:

$$\ell(\mathbf{U}, \mathbf{G}) = \sum_{i \in [n]} (-B_i + \sum_{j \in [n]} A_{ij} U_j)^2 \tag{61}$$

## C.2 Change of variables

Being able to properly normalize the input data of any neural network is a critical issue, and failing to do so can often lead to gradient explosions and other training failures (more details on data normalization in Appendix C.2). In the Poisson case study, the nodes at the boundary are constrained (*i.e.* $A_{ii} = 1$ and $A_{ij} = 0$ if $i \neq j$), and the interior nodes are not. Moreover, the coefficients of matrix $\mathbf{A}$ at these interior nodes satisfy a conservation equality (*i.e.* $A_{ii} = -\sum_{j \in [n] \setminus \{i\}} A_{ij}$). As a consequence, the distributions of their respective $B_i$ are very different, sometimes even with different orders of magnitude. It is then almost impossible to properly normalize those multimodal distribution.

In order to tackle this issue, we consider the following change of variable, changing $\mathbf{A}, \mathbf{B}$ to $\mathbf{A}', \mathbf{B}'$, and modifying the loss function accordingly. For $\mathbf{B}$, we set the dimension $d_{B'}$ of $\mathbf{B}'$ to 3 as follows:

$$B'_i = \begin{cases} [B_i, 0, 0] \text{ if node } i \text{ is not constrained} \\ [0, 1, B_i] \text{ otherwise} \end{cases} \tag{62}$$

The $B_i$'s for constrainted and unconstrainted nodes will hence be normalized independently.

Moreover, the information stored in the matrix $\mathbf{A}$ is rather redundant. As mentioned, for constrained nodes $A_{ii} = 1$ and $A_{ij} = 0$ if $i \neq j$, whereas for unconstrained nodes $A_{ii} = -\sum_{j \in [n] \setminus \{i\}} A_{ij}$. Hence the diagonal information can always be retrieved from $\mathbf{B}$ and the non diagonal elements of $\mathbf{A}$. We thus choose the following change of variable:

$$A'_{ij} = \begin{cases} A_{ij} \text{ if } i \neq j \\ 0 \text{ otherwise} \end{cases} \tag{63}$$

Finally, the loss function is transformed into the following function $\ell'$ (where $B'^p_i$ denotes the $p^{th}$ component of vector $B_i$):

$$\ell'(\mathbf{U}, \mathbf{G}') = \sum_{i \in [n]} \left( (1 - B'^2_i)(-B'^1_i) + B'^2_i (U_i - B'^3_i) + \sum_{j \in [n]} A'_{ij}(U_j - U_i) \right)^2 \tag{64}$$

One can easily check that this change of variables and of loss function defines the exact same optimization problem as in eq. (10), while allowing for an easier normalization, as well as a lighter sparse storage of $\mathbf{A}$.

## C.3 Additional super generalization experiment

This appendix describes a second experiment regarding supergeneralization. Figure 8 displays the results of the DSS model, learned without any noise, when increasing noise is added to the test examples, more and more diverging from the distribution of the training set (the graph size remains unchanged). Log-normal noise is applied to $\mathbf{A}$ ($A_{ij} \exp(\mathcal{N}(0, \tau))$), and normal noise to $\mathbf{B}$ ($B_i \mathcal{N}(1, \tau)$), for different values of noise variance $\tau$. The correlation between the results of DSS and the 'ground truth', here given by the results of LU (solving the same noisy system). But although DSS results remain highly correlated with the ground truth for small values of

Figure 8: **Increasing noise variance** $\tau$: Correlation (DSS, LU)

$\tau$, they become totally uncorrelated for large values of $\tau$ (correlation close to 0): DSS has learned something specific to the distribution $\mathcal{D}$ of linear systems coming from the discretized Poisson EDP. Further work will extend these results, analyzing in depth the specifics of the learned models.

## D  Power systems

This appendix gives more details about the AC power flow problem, and how it is converted into the DSS framework.

The AC power flow equations model the steady-state behavior of transportation power grids. They are an essential part of both real-time operation and long-term planning. A thorough overview of the domain is provided in [34].

Let's consider a power grid with $n$ nodes. The voltage at every electrical node is a sinusoid that oscillates at the same frequency. However, each node has a distinct module and phase angle. Thus, we define the complex voltage at node $i$, $V_i = |V_i|e^{\mathbf{j}\theta} \in \mathbb{C}$ (where $\mathbf{j}$ is the imaginary unit).

The admittance matrix $\mathbf{Y} = (Y_{ij})_{i,j\in[n]}; Y_{ij} \in \mathbb{C}$ defines the admittance of each power line of the network. The smaller $|Y_{ij}|$, the less nodes $i$ and $j$ are coupled. For $i, j \in [n]$, the coefficient $Y_{ij}$ models the physical characteristics of the power line between nodes $i$ and $j$ (*i.e.* materials, length, etc.).

At each node $i$, there can be power consumption (houses, factories, etc.). The real part of the power consumed is denoted by $P_{d,i}$ and the imaginary part by $Q_{d,i}$. The subscript $d$ stands for "demand". Additionally, there can also be power production (coal or nuclear power plants, etc.). They are very different from consumers, because they constrain the local voltage module. They are defined by $P_{g,i}$ and $V_{g,i}$. The subscript $g$ stands for "generation". Nodes that have a producer attached to it are called "PV buses" and are denoted by $I_{PV} \subset [n]$. The nodes that are not connected to a production are called "PQ buses" and are denoted by $I_{PQ} \subset [n]$.

Moreover, one has to make sure that the global energy is conserved. There are losses at every power line that are caused by Joule's effect. The amount of power lost to Joule's effect being a function of the voltage at each node, it cannot be known before the voltage computation itself. Thus, to make sure that the production of energy equals the consumption plus the losses caused by Joule's effect, we need to be able to increase the power production accordingly. In this work we use the common "slack bus" approach which consists in increasing the production of a single producer so that global energy conservation holds. This node is chosen beforehand and we denote it by $i_s \in [n]$.

Thus the system of equations that govern the power grid is the following:

$$\forall i \in [n] \setminus \{i_s\}, \quad P_{g,i} - P_{d,i} = \sum_{j\in[n]} |V_i||V_j|(\text{Re}(Y_{ij})\cos(\theta_i - \theta_j) + \text{Im}(Y_{ij})\sin(\theta_i - \theta_j)) \quad (65)$$

$$\forall i \in I_{PQ}, \quad -Q_{d,i} = \sum_{j\in[n]} |V_i||V_j|(\text{Re}(Y_{ij})\sin(\theta_i - \theta_j) - \text{Im}(Y_{ij})\cos(\theta_i - \theta_j)) \quad (66)$$

$$\forall i \in I_{PV}, \quad |V_i| = V_{g,i} \quad (67)$$

The encoding into our framework requires a bit of work. For the coupling matrix we use $d_A = 2$ and $A_{ij} = [\text{Re}(Y_{ij}), \text{Im}(Y_{ij})]$. For the local input we take $d_B = 5$ and $B_i = [P_{g,i} - P_{d,i}, Q_{d,i}, 1(i \in I_{PQ}), V_{g,i}, 1(i = i_s)]$. Finally, for the state variable we use $d_U = 2$ and take $U_i = [|V_i|, \theta_i]$.

Taking the squared residual of eq. (65)-(67) and taking the sum over every node, we obtain the loss of eq. (11).

## E  Further implementation details

In this section we detail the implementation details that were made to robustify the training of the DSS. None of those changes alter the properties of the architecture.

**Correction coefficient**  We introduce a parameter $\alpha$ that modifies eq. (42) in the following way:

$$H_i^k = H_i^{k-1} + \alpha \times \Psi_\theta^k(H_i^{k-1}, B_i, \phi_{\to,i}^k, \phi_{\leftarrow,i}^k, \phi_{\circlearrowleft,i}^k) \quad (68)$$

Choosing a sufficiently low value of $\alpha$, helps to keep the successive $\overline{k}$ updates at reasonably low orders of magnitude.

**Injecting existing solutions**  Depending on the problem at hand, it may be useful to initialize the predictions to some known value. This acts as an offset, that can help the training process to start not too far from the actual solutions. This offset is applied identically at every node, thus not breaking the permutation-equivariance of the architecture:

$$\widehat{U}_i^k = U_{offset} + \Xi_\theta^k(H_i^k) \tag{69}$$

For instance, in the power systems application, it is known that the voltage module is commonly around 1.0, while the voltage angle is around 0. Thus we used $U_{offset} = [1, 0]$ (keeping in mind that $d_U = 2$). On the other hand, in the linear systems application, there is no reason to use such an offset, so we used $U_{offset} = [0]$ (keeping in mind that here $d_U = 1$). But in several contexts, there exists some fast inaccurate method that can give an approximate solution closer to the final one than $(0, \ldots, 0)$.

**Data normalization**  In addition to a potential change of variables (which helps disentangle multimodal distributions of the input data, see Appendix C.2), it is also critical to normalize the input Interaction Graphto help with the training of neural networks. Each function $\Phi_{\rightarrow,\theta}^k$, $\Phi_{\leftarrow,\theta}^k$ and $\Phi_{\circlearrowright,\theta}^k$ take $A_{ij}$ as input, and the functions $\Psi_\theta^k$ take $b_i$ as input. We thus introduce hyperparameters $\mu_A, \sigma_A \in \mathbb{R}^{d_A}$ and $\mu_B, \sigma_B \in \mathbb{R}^{d_B}$ are used to create a normalized version of the data:

$$a_{ij} = \frac{A_{ij} - \mu_A}{\sigma_A} \tag{70}$$

$$b_i = \frac{B_i - \mu_B}{\sigma_B} \tag{71}$$

$\mathbf{g} = (\mathbf{a}, \mathbf{b})$ (with $\mathbf{a} = (a_{ij})_{i,j \in [n]}$ and $\mathbf{b} = (b_i)_{i \in [n]}$) is thus the normalized version of $\mathbf{G}$. We apply the DSS to this normalized $\mathbf{g}$ and consider the loss $\ell(Solver_\theta(\mathbf{g}), \mathbf{G})$ instead of $\ell(Solver_\theta(\mathbf{G}), \mathbf{G})$.

**Gradient clipping**  We sometimes observed (e.g., in the power systems experiments) some gradient explosions. The solution we are currently using is to perform some gradient clipping. Further work should focus on facilitating this training process automatically.