[Reviews · NeurIPS 2020]

Review 1

Summary and Contributions: This paper introduces a novel class of solvers (Deep Statistical Solvers) that: (a) are based on learning message-passing functions on a Graphical Neural Network and (b) are aimed at predicting the solutions of general (linear or nonlinear) equations defined on these graphs via a specified permutation-invariant loss function. Instead of using pre-calculated numerical solutions for each sampled problem as groundtruth, the method optimizes the loss function directly. In Section 4, the authors discuss important theoretical results showing the existence of accurate solvers within a specific class of considered models. They later apply their method to two practical problems (2D Poisson equation solver and AC power flow calculation) and show its effectiveness.

Strengths: The problem that this paper is addressing is defined rigorously and clearly. Theoretical proof of the existence of the solver in a particular family of models (Theorem 1 and Corollary 1) is interesting and important in the context of this work. It appears to be sound and is well explained (although I have not carefully checked a more detailed discussion in the supplementary materials). While it would be interesting to know more about the construction of distributions D that satisfy required hypotheses and conditions of uniqueness and continuity of U with respect to G, these questions do not appear to be critical in the context of this paper. The empirical evaluation is also sound with a minor exception of a few questions mentioned below. I believe this work to be novel and significant in the context of machine-learning inspired approaches to predicting solutions of complex (frequently physically motivated) equations.

Weaknesses: While overall I believe the paper to be of a high quality, there were a few minor issues that I wanted to emphasize: 1. Corollary 1 is an interesting and useful claim in itself. However, it relies on uniqueness and continuity of a minimizer U with respect to G. The paper provides two experimental setups, but does not seem to hint at whether these two conditions are actually expected to be satisfied in these cases. Naively, I would expect that both conditions might be satisfied for a discretized Poisson equation. However, in a highly nonlinear AC power flow computation both of them may be expected to fail in certain cases. Even a sentence about this could be insightful in the light of the effort that went into proving Theorem 1. 2. It would be useful to have an explicit definition of the correlation that was actually used to arrive at 99.99% number in the main text. Without it, I find it difficult to confidently judge the corresponding raw in Table 1 and afterwards. 3. I am a little confused by the sentence on line 283 saying "... keep in mind that the main goal is to predict power flows, and not voltages per se". While accurately predicting P and Q may have more important practical implications, correctness of V and \theta laying at the heart of the optimization objective seems to be a primary goal from the point of view of properly solving the original equation. In this light, any speedup at the cost of a poor prediction accuracy of V and \theta appears to be meaningful at the very least from the mathematical point of view. This seems to be reflected in Table 2 that seems to indicate a very high correlation (which I assume is for P and Q and not for V and \theta, something worth mentioning in the table itself), but quite large losses compared to those of NR.

Correctness: After reviewing theoretical claims and empirical methodology, I believe them to be correct. There were a few minor issues that stood out to me that I outlined above.

Clarity: The paper is well written and overall sufficiently clear. There are just a few minor issues (like "remaining" -> "remainder" on line 87) that are worth correcting in the final version of the text.

Relation to Prior Work: The prior work is not discussed explicitly in a separate section, but is instead mentioned throughout the text. I also noticed that this paper does not always cite related prior work. For example, the "proxy-less" method discussed in the lines 107-121 and illustrated in Figure 2 has been used before in other similar articles aimed at predicting solutions of PDEs. I believe "DGM: A deep learning algorithm for solving partial differential equations" to be one of the examples of such prior work that would be worth mentioning here.

Reproducibility: Yes

Additional Feedback:


Review 2

Summary and Contributions: The paper introduces a new approach for approximating solutions of nonlinear systems (or optimization problems) based on graph neural nets. Recent work in ML approximation of solvers simply assembled a large set of input-output pairs, and used supervised learning to train predictive model to extrapolate to unseen inputs. The approach in this paper instead directly looks at the objective function (or rather a large number of samples from a distribution of objectives), and attempts to learn a parameterized solver to find solutions for problems from this distribution. Note that this isn't simply stochastic gradient descent applied to optimize the objective function, but rather the authors train a parameterized graph-neural net based on many samples from a distribution of objective functions, that can "look-up" the solutions at its inference step.

Strengths: Overall this direction is an interesting development compared to the recent literature -- in that the goal is to explicitly find solutions that minimize the objective function rather than approximating the solution vectors. This allows to take into better account the problem domain (the "physics" of the problem) -- rather than use the original system as a black-box. Experimental results show that the approach is promising for the two scenarios considered (although arguably this wasn't a thorough test). The paper also makes a theoretical contribution by showing that DSS, the class of such parameterized graph-neural nets, is a universal approximator of certain permutation-invariant (and equivariant) functions, following the proof steps of a recent GNN paper.

Weaknesses: While I find the proposed approach interesting and promising, I find the claims too strong (I would expect a more refined discussion on what type of problems, and what additional assumptions are needed for DSS to perform well) and the experimental validation lacking, and some of the simple baselines likely misrepresented. In general, replacing known robust solutions for some critical domains (e.g. power-flow, aerodynamics simulations for airplanes, e.t.c) by fast approximate solvers is exciting, but also dangerous -- and to what extent you can guarantee that DSS can do well for the more difficult cases? Is there a possibility of catastrophic failures? Are there applications you envision where this risk is reduced? 1) While the authors establish "universality" -- similar to the corresponding results for plain feed-forward NNs, there's no way to ensure that the GNN will actually be able to find the best function in that space. So we have to rely on extensive experimental validation to test it. In particular I imagine that DSS works much better for some types of problems (e.g. easy smoothing-type of problems), and much worse for "frustrated" or ill-conditioned systems, having a very difficult time to match direct solutions. I would like to see some discussion around limitations of the approach, and perhaps additional assumptions that you're implicitly imposing. 2) The case of Poisson equation leading to a simple linear system -- is a nice "sanity check", but given that the paper is addressing a general class of nonlinear systems, PDEs, and optimization problems, I would expect a broader array of test cases. a) The baseline numbers for plain linear-system solvers seem dramatically slow: in my simple tests on random asymmetric dense linear systems using python numpy.linalg.solve for 500x500 matrix, it requires a running time of around 3-4ms , whereas the paper claims LU takes about 2 seconds. (An order of 1000x off!) Are the units correct? In 2 seconds one can solve linear systems of size 5000x5000. b) For iterative solvers, the number of iterations to reach a given level of accuracy depends on the condition number of the system. The GNN message-passing solution is more akin to an iterative solver so probably behaves similarly. So I'd be curious if the fixed number-of-steps approach makes sense, and works for more challenging problems. In particular -- interesting practical distributions of problems will have both high and low condition numbers, so adaptive number of steps (to reach a given tolerance) would make more sense. For example this would hold true for the linear Gaussian belief propagation (a naive version of a GNN). c) Also the Poisson equation has a very small number of unique parameters, and it would be interesting to see if a GNN is able to learn how to solve non-homogeneous versions of linear systems, with many more unique parameters. 3) The paper does have a nonlinear system arising from the Power-flow problem. However, based on particular inputs, power-flow can exhibit a very easy linear-like behavior, but also a very complex nonlinear behavior when the system is under a heavy load / stress. Taking random samples of loads it's likely you're mostly in the easy linear part, which is easy to solve, and hence you get very high correlations. I would like to see how well the DSS approach behaves as you explicitly try to come to the boundary of feasibility -- e.g. by increasing the loads. I would be also interested to look not at correlations, but at the worst-error over the data. A collection of realistic test-cases for power-flow is available in "The Power Grid Library for Benchmarking AC Optimal Power Flow Algorithms", by Carleton Coffrin et.al. 4) Are there any guarantees that the GNN solution will satisfy the standard OPF feasibility constraints? especially in the more difficult cases?

Correctness: I find the experiments lacking -- but I didn't see any errors (apart from the very slow running times for linear solvers). I skimmed, but did not verify the theoretical results on universal approximation thoroughly. They make sense, and follow a related recent result for GNNs. Furthermore, k-steps of message passing can be viewed as a forward pass in a computation tree for the graph, (like unrolling the messages in belief propagation into a large tree with local structure repeating the structure of the graph). Hence the results on universal approximation from feed-forward DNNs may be applicable for the computation tree, and extend to GNNs.

Clarity: The paper is mostly well-written, although some notation can be improved.

Relation to Prior Work: The paper has good references to recent work. However, there should be some mention of the graphical-models literature, which seems directly relevant. There is a long tradition of defining energy functions composed of local terms respecting a graph, and using iterative algorithms like belief propagation, for example "Gaussian belief propagation solver for systems of linear equations", by Shental, Siegel, Wolf, e.t.c. In the graphical modeling viewpoint -- learning the parameters of the graphical model acts like fitting the DSS, and doing belief propagation acts like doing inference in the corresponding GNN for optimal theta. Some pointers and comparison to that literature would be useful. By virtue of being defined w.r.t to graph, and having local potentials living on vertices and edges of the graph, they trivially satisfy permutation invariance.

Reproducibility: Yes

Additional Feedback: Overall - I think the proposed approach is a novel and interesting contribution, but the claims are too strong and not refined, with very limited and biased (easy) experimental evidence. Showing only examples where the methods works extremely well (99.99% correlation) may mislead the readers to think the method is broadly applicable. I would like a more nuanced discussion, and some examples of harder problems, where the method doesn't work as well. Additional comments: 1) You mention not relying on the set of training examples as an advantage. However, you still need a sample of problems to approximate the expectation of losses. How many such samples do you need compared to the supervised learning methods? 2) Some notation in section 2.1 is not clear. R^d_A and R^d_B is undefined and confusing. Reading through I can see this is just a vector of the possible edges, and vertices. In setion 3, M_theta^k is used before being defined. 3) The only assumptions on the loss is continuity and permutation invariance. Are there other assumptions that make it more likely that DSS will work? 4) k and d should be chosen based on distribution D. Vague. Any more insight? Can you get away with fixed k and d? 5) "Hyper-parameters not explicitly mentioned are found by trial and error". It'd be good to at least include the list, and the range of the parameter grid explored. Thank you for the feedback. I would have preferred if you agreed to include current results on more difficult cases (the ongoing work you mentioned) in the paper -- it's fine if it doesn't work great everywhere, as long as its promising in some settings. However, only showing results where it works really well without further discussion could mislead some readers. I increase my rating a bit -- I do hope that the discussion about limitations of the approach in the hard cases is very clearly reflected in the revised paper, and suggest / explore to use it as a preconditioner for a technique with guarantees in critical applications.


Review 3

Summary and Contributions: *** Post rebuttal update *** I have read the authors' response and am satisfied with way they addressed the points raised and the modifications they'll introduce based on my feedback. As a result my score remains unchanged. ********************************************** The paper introduces Deep Statistical Solvers (DSS), a new class of trainable solvers for optimization problems, arising from system simulations and others. The key idea is to learn a solver that generalizes to a given distribution of problem instances. This is achieved by directly using as loss the objective function of the problem, as opposed to most previous which mimic the solutions attained by an existing solver. The paper is framed as a methodological contribution and the authors also provide empirical evaluation of their proposed solver.

Strengths: The paper is framed as a methodological contribution and while the general idea introduced in the paper has merit, some of the details, as described in the paper, and the implementation are a bit concerning to me.

Weaknesses: Some itemized concerns below: -- The paper frames the use of the proxy problematic due to the need for labelled data however the proposed approach also relies on labelled data. -- The performance with classical methods is performed in an odd way: what is "Correlation with LU"? Typically in numerical analysis ones compares the two methods in terms of RMSE of the solution or the approximation error (e.g. in terms of Frob norm). -- Comparison in terms of computational complexity is needed, i.e. wall clock time or flops given the same model parallelism. -- Figure 2 is not very informative. While the proposed method does not use a 'proxy' their model presumably encodes some of the constraints (geometry of the problem, boundary constraints) in terms of the network architecture they used for training. Hence, even though there is no proxy per se, the network engineering amounts to having a proxy. --First example in Sec 5.1 what do A and B represent in the discretization problem involving the house?

Correctness: I did not find inaccurate claims, although there are some confusing details highlighted in the "Weaknesses" section above.

Clarity: Overall the paper is clearly written and overall it meets the bar in terms of clarity, yet there are some generalizations throughout the paper (some comments in the Weaknesses section above). -- The use of the term 'bricks' in line 21 is a bit odd.

Relation to Prior Work: The paper discusses prior work in multiple fields, from numerical analysis to deep learning. It cites relevant lines of work and related methods, as needed.

Reproducibility: Yes

Additional Feedback:


Review 4

Summary and Contributions: This paper discusses a new machine learning approach that replaces numerical methods-based computational blocks with a model-based approach. Unlike prior work, however, this model can be trained by directly minimizing the cost function of interest, which bypasses the (sometimes costly) requirement of obtaining many input-output pairs for each problem. A framework called Statistical Solver Problem (SSP) is developed which solves numerical optimization problems containing permutation-invariant variables by training a model to produce the system state that minimizes the objective function. Graph neural networks are used to implement the model. A few proofs are provided which show that this framework is able to approximate the problems of interest to an arbitrary degree of accuracy. Experiments are provided for two domains, consisting of linear systems described from discretized PDEs and AC power flows.

Strengths: The presented framework is very well described, and its formulation is presented in a general manner such that it can be applied to a variety of problems. The motivation is also very clear: problem instances are generally far easier to obtain than their solutions, and therefore the ability to train a model without solutions is great to have. Furthermore, the experimental results demonstrate that the method is able to provide reasonable solutions in a much faster time than numerical methods-based approaches.

Weaknesses: There are two primary weaknesses of this work from my perspective: -The introduction discusses problems which have constraints and the importance of producing solutions which satisfy these constraints; however, the presentation of the method does not discuss constraints at all. Since the approach requires the loss be differentiable (and require a useful gradient signal), this method does not seem to me like it can be guaranteed to produce valid solutions. Instead, the best one can do is to add a constraint violation penalty to the loss and hope the produced solutions are valid (this is done in the power flow experiments). -A primary motivation of this paper is the fact that it may be expensive to obtain input-output pairs to use for training methods requiring them, which is sensible. However, no experiments are provided which compare the quality of the models produced here to prior work which trains on input-output pairs. For problems where it is feasible to obtain this type of data, it is important that we know whether this approach still outperforms other methods or if there's a gap that will need to be closed with future work.

Correctness: As far as I am aware, the presented methodology is sound.

Clarity: The paper is written with exceptional quality; the developed framework is presented in a thorough and easy-to-follow manner, and every piece is well-motivated.

Relation to Prior Work: The primary motivation driving this work - developing models which can replace more costly numerical methods-based computational blocks while not requiring input-output pairs for training - is clear, as well as how this approach differs from prior work. However, as I mentioned previously, it is unclear whether this approach is better or worse than other approaches when the correct data is available, since no comparisons are made.

Reproducibility: Yes

Additional Feedback: I think this work is very interesting and well-written, and I'm generally leaning towards acceptance. However, the approach really needs to be compared against others in order to understand how well it truly functions. --------------------------------------------------------------------------------------------------------------- POST AUTHOR RESPONSE EDIT: Thanks for your response. Since you were not able to get additional comparisons done in time for the rebuttal, I'm leaving my score unchanged. I'm still leaning towards acceptance.

[Author Response · NeurIPS 2020]

We first would like to thank the reviewers for their insightful comments and suggestions. We will give a list of changes that we propose to implement to improve the quality of the paper, based on the four reviews.

*Foreword*: This paper is framed as a methodological and theoretical contribution, with simple experimental validation. There is in particular no specific ethical concern with this paper – something we will add in a "Broader Impact" section.

**Reviewer 1** • *Corollary 1*: Continuity and the permutation invariance conditions are satisfied in both the linear system and the AC power flow problems. The uniqueness is only proven for the linear system. However it is very likely to hold for many problems. We will add one sentence along this line. • *Correlation with 'ground truth'*: RMSE and correlation (between DSS and 'exact' solutions) provide complementary views. In this context we used correlation, we will add RMSE results in supplementary material. • *AC power flow discussion*: Indeed, a large loss does not necessarily imply poor predictions in $V$ and $\theta$, nor, in turn, in $P_{ij}$ and $Q_{ij}$. But the approach remains beneficial in terms of computational cost as long as the predictions for $P_{ij}$ and $Q_{ij}$ are good enough. This will be better explained in the final version. We will also modify the caption of Table 2. • *"DGM" reference*: We will add it to references [6]-[7].

**Reviewer 2** • *Critical applications*: Our primary application domain being the highly sensitive area of power grids, we are truly aware that AI can bring novel threats to systems security and reliability. This was our primary motivation for properly introducing mathematical concepts and ideas in this work. • *Limitations*: We are aware that the universal approximation theorem that we derive does not offer any guarantee of convergence. We will revise the paper to stress this more clearly. However, this non-trivial result is a pre-requisite to provide solid theoretical ground to the proposed approach. • *Experimental validation*: Adding more difficult use cases (pushing the loads to the limit in the power flow, handling non-homogeneous and even nonlinear PDEs as suggested, etc.) is on-going work. We will more clearly discuss the current limitations in the conclusion. In particular, we will mention the fact that in the case of critical applications, our approach can be used to find a good starting point for some classical method, thus saving large amounts of computing resources. • *CPU time*: We probably did not present our results clearly enough, but the unit used for CPU times in Table 1 is ms, making our estimation in line with the one from the review. We will modify the row and column captions. • *Adaptive number of iterations*: During our preliminary experiments we observed that having different neural network blocks at each propagation step gave better results, thus making $\overline{k}$ a fixed hyperparameter. However, a recurrent graph neural network structure with adaptive $\overline{k}$ is definitely an interesting avenue to explore. • *AC linear domain*: This aspect has been further investigated in some of our previous work, which we will cite in the revised version, by comparing our statistical solver, the Newton-Raphson method, and the DC-approximation (which relies on a linearization of the Kirchhoff's equations). These experiments showed that the DC-approximation was significantly worse than our proposed method, thus proving that the DSS performs well in the non linear domain of the AC PF problem. We acknowledge that this aspect should have been addressed in the paper. • *AC feasibility*: Our architecture takes as input the active and reactive loads, active production and voltage setpoints at generators, and we use a slack bus so the AC feasibility is not an issue. • *References*: We were not aware of the Shental et al. reference, and will add it in the discussion. • *Amount of samples*: We do need some sample problems, but we do not need to have their solutions, as in the supervised "proxy" approach, since we have a closed-form expression for the loss function. We will stress this better in the revised version. Nevertheless, it is true that the choice of the problems (i.e. of the distribution $\mathcal{D}$) is another important issue regarding the validation. However, the same issue arises in the proxy case. • *Notations*: $\mathbb{R}^{d_A}$ and $\mathbb{R}^{d_B}$ are introduced on line 138 and $M_\theta^k$ on line line 137, before being used in eq. (8). We will regroup clear notation definitions in the revised version for additional clarity. • *Hypotheses*: In addition to the continuity and permutation invariance hypotheses, we also need to have some uniqueness property as introduced in Section III to be able to properly lay the ground for the theorem. • *Choice of $\overline{k}$ and $d$*: If $\mathcal{D}$ is a dataset of snapshots of the Californian power grid, then one can compute the power grid diameter, which we proved to be a good lower bound for $\overline{k}$. Moreover, one can expect that the larger the inputs $d_A$ or $d_B$, the larger $d$ should be. We will add a sentence in the revised version. • *Hyperparameters*: We will add the ranges of the grid search for all hyperparameters.

**Reviewer 3** • *Labelled data and 'proxy' approach*: We use the word 'proxy' to designate the method that learns to reproduce existing solutions. Here, we only need some sample problems, but not their solutions, thanks to the closed-form expression for the loss function. We will stress, and phrase, this better in the revised version. • *Correlation with 'ground truth'*: See response to reviewer 1. • *Computational complexity*: The theoretical complexity has been addressed line 152. The wall-clock times have been reported in Table 1 and Table 2 for all considered methods. • *Meaning of* **A** *and* **B**: $A$ and $B$ are the matrix (resp. vector) that result from the discretization and assembling process of the Poisson equation. The chosen loss amounts to solve $AU = B$. Each element of $U$ corresponds to a node that is shown in Figure (4). Moreover, the edges that are displayed at the top left correspond to non zero values $A_{ij}$. Further details are provided in lines 75-77 and Appendix A. • *Phrasing*: The term "brick" is employed to express the idea that neural networks are here elementary building blocks in a broader architecture. We will modify the wording.

**Reviewer 4** • *Constraints*: The term "constraint" is used in the introduction: it does not refer to optimization constraints but to domain specific consideration. As defined in eq. (2)-(3), we focus on unconstrained optimization. Adding non-trivial constraints is actually a line of work we are currently investigating. We will correct this misleading phrasing. • *Comparison with proxy solutions*: We fully agree regarding the lack of comparison with proxy neural networks. This is on-going work.

[Meta-Review · NeurIPS 2020]

The paper proposes new theoretical results regarding universal approximation property of graph convolutional neural networks and uses and trains them for (approximately) solving optimization problems defined on graphs, in particular arising from a discretization of PDEs. The solver is trained directly from the model energy. The paper was recognized by reviewers as having an interesting contribution and meeting the quality standards. The authors are invited to submit the final version including the rebuttal points, addressing all minor revision issues and the literature connections mentioned. Showing the applicability boundaries by studying failure cases is also highly appreciated. Some additional comments from discussion: The applicability of the method seems quite limited the model must be already known (not learned, in contrast to common way of applying ML). Interestingly, the work demonstrates solving problems with constraints by introducing constraints as penalties during learning. In this case it would be a natural concern that for more complex problems the learning will be dominated by trying to satisfy the constraints, as the cost is taken in the expectation. Generalization to other topologies for problems harder that just smoothing has also not been shown. Perhaps the method may find more trustworthy applications in learning preconditioners or starting solutions for existing solvers with guarantees. Simple fixed point iteration solvers may look very similar to GNN. Learning a preconditioner would have advantage of keeping guarantees of a solver while speeding it up on average. On the literature side, the idea is not completely novel. In vision, unsupervised stereo and optical flow neural networks have been learned from a hand-engineered total variation cost models (trying to minimize the cost, not knowing the optimal solution). In [Vogel and Pock: "A Primal Dual Network for Low-Level Vision Problems" (2017) Appendix Section 3] the NN is trained with the objective to minimize the ROF functional for image restoration. These networks are not graph-based, but nevertheless, this key idea has been used before. On the side of message-passing algorithms like belief propagation, there've been many works training generalized solvers in a supervised setting. I.e. learning the inference (solver) directly, bypassing the model.